# Cell-surface tethered promiscuous biotinylators enable comparative small-scale surface proteomic analysis of human extracellular vesicles and cells

Lisa L Kirkemo[1†], Susanna K Elledge[1†], Jiuling Yang[2,3], James R Byrnes[1], Jeff E Glasgow[1], Robert Blelloch[2,3], James A Wells[1,4]*

[1]Department of Pharmaceutical Chemistry, University of California, San Francisco, San Francisco, United States; [2]Department of Urology, University of California, San Francisco, San Francisco, United States; [3]The Eli and Edythe Broad Center of Regeneration Medicine and Stem Cell Research, University of California, San Francisco, San Francisco, United States; [4]Department of Cellular and Molecular Pharmacology, University of California, San Francisco, San Francisco, United States

*For correspondence:
jim.wells@ucsf.edu

†These authors contributed equally to this work

Competing interest: The authors declare that no competing interests exist.

**Abstract** Characterization of cell surface proteome differences between cancer and healthy cells is a valuable approach for the identification of novel diagnostic and therapeutic targets. However, selective sampling of surface proteins for proteomics requires large samples (>10e6 cells) and long labeling times. These limitations preclude analysis of material-limited biological samples or the capture of rapid surface proteomic changes. Here, we present two labeling approaches to tether exogenous peroxidases (APEX2 and HRP) directly to cells, enabling rapid, small-scale cell surface biotinylation without the need to engineer cells. We used a novel lipidated DNA-tethered APEX2 (DNA-APEX2), which upon addition to cells promoted cell agnostic membrane-proximal labeling. Alternatively, we employed horseradish peroxidase (HRP) fused to the glycan-binding domain of wheat germ agglutinin (WGA-HRP). This approach yielded a rapid and commercially inexpensive means to directly label cells containing common N-Acetylglucosamine (GlcNAc) and sialic acid glycans on their surface. The facile WGA-HRP method permitted high surface coverage of cellular samples and enabled the first comparative surface proteome characterization of cells and cell-derived small extracellular vesicles (EVs), leading to the robust quantification of 953 cell and EV surface annotated proteins. We identified a newly recognized subset of EV-enriched markers, as well as proteins that are uniquely upregulated on Myc oncogene-transformed prostate cancer EVs. These two cell-tethered enzyme surface biotinylation approaches are highly advantageous for rapidly and directly labeling surface proteins across a range of material-limited sample types.

## Editor's evaluation

This work presents useful technical options in the detection and comparison of the cell surface and extracellular vesicle (EV) proteomes of normal and myc-transformed cells in culture. The procedures and the proteomes identified show promise of wider application in clinical and basic research settings. The work highlights distinctions in the two proteomes and the influence of myc-transformation on the selection of cargo molecules for capture in EVs secreted by human cells in vitro.

**Table 1.** Current methods available for cell surface biotinylation.

| Method | Protocol length (time) | Selectivity | Sample size requirement |
|---|---|---|---|
| Biocytin hydrazide | +++ | +++ | +++ |
| Sulfo-NHS-LC-LC-biotin | ++ | + | + |
| APEX2/HRP | + | + | + |

## Introduction

The cell surface proteome, termed the surfaceome, serves as the main communication hub between a cell and the extracellular environment (*Wollscheid et al., 2009*). As such, this cellular compartment often reveals the first signs of cellular distress and disease, and is of substantial interest to the medical community for diagnostic and therapeutic development (*Leth-Larsen et al., 2010*). The precise and comprehensive profiling of the surfaceome, termed surfaceomics, provides critical insights for our overall understanding of human health and can inform drug development efforts. Several strategies have emerged for either selective or comprehensive surfaceomics, including biocytin hydrazide labeling of surface glycoproteins (*Wollscheid et al., 2009*), chemical biotinylation of lysines via NHS-ester labeling (*Huang, 2012*), and promiscuous biotinylator fusion proteins (APEX2, BioID, and SPPLAT) (*Rees et al., 2015*; *Sears et al., 2019*; *Wollscheid et al., 2009*). Membrane protein enrichment is a necessary step in surfaceomics, due to the inherent low abundance of membrane proteins compared to cytosolic proteins, and their identification can be overwhelmed by cytosolic contaminants. While each of these strategies robustly label surface proteins, they: (1) require large sample inputs (biocytin hydrazide), (2) require production of genetically engineered cells (APEX2 and BioID), (3) label only partner proteins by binding targeting antibodies fused to APEX2 or horseradish peroxidase (HRP; SPPLAT), (4) require extensive sample manipulation (biocytin hydrazide), or (5) exhibit increased nonspecific labeling (NHS-ester, *Table 1*; *Bausch-Fluck et al., 2012*; *Elschenbroich et al., 2010*; *Griffin and Schnitzer, 2011*; *Kuhlmann et al., 2018*; *Li et al., 2020b*). Moreover, many of these methods are not able to capture short and transient changes that occur at the cell surface, such as binding, adhesion, assembly, and signaling (*Kalxdorf et al., 2017*). These current methods complicate the direct characterization of small clinical samples such as extracellular vesicles (EVs) in patient serum. As biological research increasingly depends on animal models and patient-derived samples, the requirement for simple and robust methods amenable to direct labeling of material-limited samples for proteomic analysis will become paramount.

Exosomes and other small EVs are produced by both healthy and diseased cells (*Colombo et al., 2014*). In cancer, these small EVs contribute to tumor growth and metastasis, modulate the immune response, and mediate treatment resistance (*Al-Nedawi et al., 2008*; *Edgar, 2016*; *Kalluri and LeBleu, 2020*; *Shurtleff et al., 2018*). Consequently, these EVs are a focus of intense clinical investigation. Recent studies suggest that small EVs incorporate proteins and RNA from the parent tumor from which they originate (*Lin et al., 2015*; *Soung et al., 2017*), and certain proteins may be preferentially shuttled into EVs (*Poggio et al., 2019*). There is also strong evidence that cancer-derived EVs are unique from the EVs derived from healthy surrounding tissues, and therefore represent a promising target for noninvasive, early detection diagnostics, or EV-focused therapies (*Kalluri and LeBleu, 2020*; *Skog et al., 2008*; *Zhou et al., 2020*). However, strategies for the unbiased profiling of small EV membrane proteomes remain limited. Isolation of high-quality, enriched small EV populations is challenging, requiring numerous centrifugation steps and a final sucrose gradient isolation, precluding the use of current labeling methods for membrane proteome characterization (*Poggio et al., 2019*; *Shurtleff et al., 2018*). Strategies to characterize the surface proteome of small EVs would propel biomarker discovery and enable the differential characterization of small EVs from that of the parent cell. These important studies could help illuminate mechanisms underlying preferential protein shuttling to different extracellular vesicle populations.

Here, we functionalize the promiscuous biotinylators, APEX2 and HRP, as noncellularly encoded exogenous membrane tethering reagents for small-scale surfaceomics, requiring <5e5 cells. This method is 10- to 100-fold more rapid than other existing protocols and requires fewer wash steps with less sample loss. Likewise, due to its selectivity toward tyrosines, it is not hindered by variability in individual protein glycosylation status (*Leth-Larsen et al., 2010*) or by impeding complete tryptic

peptide cleavage through modification of lysines (*Hacker et al., 2017*), like biocytin hydrazide or NHS-biotin methods, respectively. Using this robust new strategy, we performed surfaceomics on cells and corresponding small EVs from a cellular model of prostate cancer using the prostate epithelial cell line, RWPE-1 with or without oncogenic Myc induction. While certain proteins show increased expression in both parental cell and EV surfaces, a subset of proteins was found to be either pan-EV markers (ITIH4, MFGE8, TF, DSG1, TSPAN14, AZGP1, and IGSF8) or selectively enriched with Myc overexpression in cancer-derived EVs (ANPEP, SLC38A5, FN1, SFRP1, CDH13, THBS1, and CD44). These differentially regulated proteins pose interesting questions related to preferential protein shuttling, and the proteins upregulated in both cellular and EV contexts reveal candidates for early stage urine or serum-based detection without invasive surgical intervention. We believe these simple, rapid, and direct surfaceomic labeling tools may be broadly applied to small-scale surfaceomics on primary tissues.

## Results

### Generation of promiscuous cell-surface tethered peroxidases for exogenous addition to cells

Both APEX2 and HRP are broadly used promiscuous proximity biotinylators that label nearby tyrosine residues in proteins through a radical intermediate mechanism using a biotin-tyramide reagent (*Figure 1A*; *Hung et al., 2016*; *Martell et al., 2016*). HRP has been targeted to specific cell-surface proteins through antibody conjugation to label target proteins and their binding partners (*Rees et al., 2015*). More recently, HRP was used as a soluble cell surface labeler to identify rapid cell surface proteome changes in response to insulin (*Li et al., 2021*). Genetically encoded, membrane-targeted APEX2 and HRP have also permitted promiscuous labeling of proteins in specific cellular compartments, but these efforts required cellular engineering (*Hung et al., 2016*; *Li et al., 2020a*). We sought to expand the use of these tools to biotinylate surface proteins of cells without the need for cellular engineering, enabling the specific enrichment of surface-resident proteins for mass spectrometry analysis.

The first approach we tested was to tether a DNA-APEX2 conjugate to the cell membrane through a lipidated DNA anchor. Gartner et al. have shown lipidated DNA anchors can tether together molecules or even cells (*McGinnis et al., 2019*; *Weber et al., 2014*). Here, the lipidated DNA is first added to cells, then hybridized with a complimentary strand of DNA conjugated to APEX2 (*Figure 1B*, left panel). To conjugate DNA to APEX2, we leveraged the single unpaired cysteine in the protein for site-specific bioconjugation of the complementary DNA. We first reacted APEX2 with DBCO-maleimide, after which the DBCO moiety was readily conjugated with azido-DNA. The kinetics of coupling was monitored using LC-MS and the conjugate was purified by nickel column chromatography, yielding a single conjugated product (*Figure 1—figure supplement 2A*) that retained full enzymatic function relative to unlabeled APEX2 (*Figure 1—figure supplement 2B*). Microscopy was used to observe the colocalization of DNA-conjugated APEX2 to the membrane (*Figure 1C*). This result was recapitulated using flow cytometry, indicating that this approach results in surface tethering of APEX2, an important step toward the specific labeling of the cell surfaceome (*Figure 1—figure supplement 2C*).

To avoid the need for bioconjugation, we also tested a commercially available reagent where the promiscuous biotinylator HRP is conjugated to the lectin wheat germ-agglutinin (WGA) (*Figure 1B*, right panel). WGA-HRP is used regularly in the glycobiology and neuroscience fields to label cell membranes for immuno-histochemistry and live-cell imaging (*Mathiasen et al., 2017*; *Wang and Miller, 2016*). This is an inexpensive and widely available tool that only requires the presence of surface protein N-acetylglucosamine (GlcNAc) and sialic acid glycans to localize HRP to the membrane. The successful and rapid colocalization of WGA-HRP to the plasma membrane (PM) compared to HRP alone was verified using immunocytochemistry, indicating this approach is a potential alternative for cell surface labeling (*Figure 1D*). Further testing showed that adding WGA-HRP to cells in the presence of hydrogen peroxide and biotin-tyramide led to robust surface labeling even with no preincubation time (*Figure 1—figure supplement 3*).

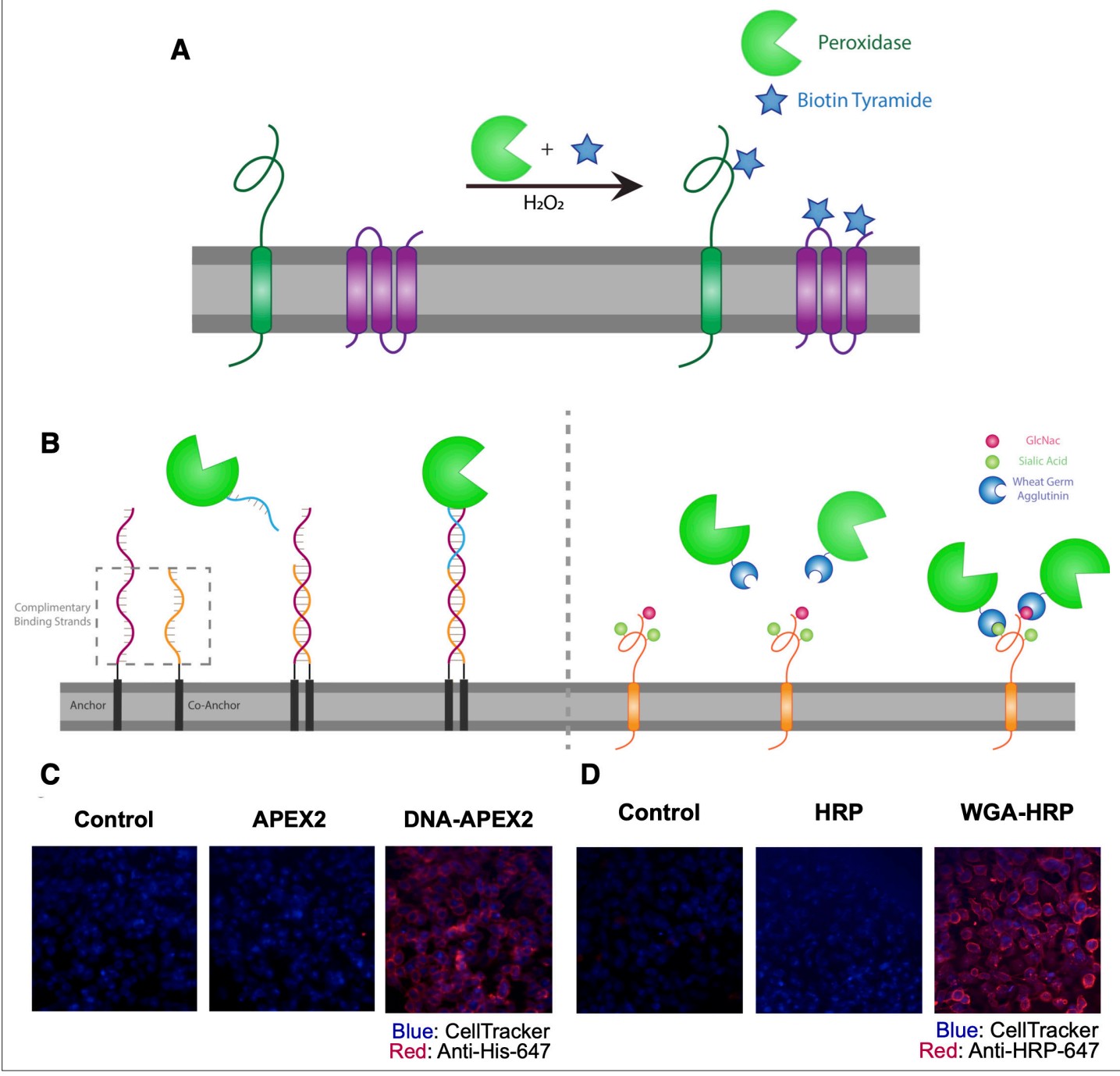

**Figure 1.** Direct labeling of promiscuous biotinylators to the cell membrane for rapid cell surface proteome characterization of small-scale biological samples. (**A**) Outline of enzymatic reaction mechanism. APEX2 and HRP both require biotin tyramide and hydrogen peroxide to produce the biotin-radical intermediate. (**B**) Tethering either enzyme is completed through differing mechanisms: (i) APEX2 is tethered through bio-conjugation of a single-strand of DNA, which is complementary to an exogenously added sequence of lipidated-DNA attached to the membrane, (ii) Commercially available wheat germ agglutinin (WGA)-HRP associates with native GlcNAc and sialic acid glycan moieties on cell surface proteins. (**C**) Microscopy images depicting the localization of DNA-APEX2 to the cell surface of KP-4 cells after the introduction of the lipidated-DNA complementary strands. (**D**) Microscopy images depicting the localization of WGA-HRP to the membrane of KP-4 cells. All microscopy images are representative of two biological replicates.

The online version of this article includes the following figure supplement(s) for figure 1:

**Figure supplement 1.** Expression, purification, and validation of APEX2 enzyme.

**Figure supplement 2.** Labeling and efficacy of APEX2 with DNA.

**Figure supplement 3.** WGA-HRP preincubation time on cells has no effect on labeling efficiency.

## Cell-tethered biotinylators more effectively label the surfaceome than nontethered biotinylators and are comparable to biocytin hydrazide

Next, we set out to optimize labeling conditions for small-scale sample characterization. As APEX2 is kinetically slower than HRP (*Lam et al., 2015*), we used APEX2 to establish a suitable concentration range of enzyme for cell surface labeling. We found that 0.5 µM APEX2 produced maximal labeling of cells (*Figure 2—figure supplement 1A*) and maintained equivalent labeling across a range of cell numbers (2.5e5–1e6 cells; *Figure 2—figure supplement 1B*). Next, we compared the efficiency of DNA-APEX2, WGA-HRP, and their nontethered counterparts to biotinylate a small sample of 5e5 Expi293 cells. We found a 5- to 10-fold increase in biotin labeling for both tethered DNA-APEX2 and WGA-HRP relative to nontethered controls as assessed by flow cytometry (*Figure 2A*) and western blotting (*Figure 2B*). Moreover, tethered DNA-APEX2 and WGA-HRP systems exhibited similar biotinylation efficiency, suggesting either system is suitable for small-scale surfaceomics. Having both systems is useful, as some cells may not widely express glycoproteins recognized by commercially available lectin-HRP conjugates—such as some prokaryotic species—and therefore could require the glycan-agnostic DNA-tethered APEX2 construct (*Schäffer and Messner, 2017*).

To compare the degree of surface protein enrichment these two systems offer, we enriched biotinylated proteins generated with either approach and compared the resulting enrichments using LC-MS/MS. As an initial efficacy comparison, cell surface labeling with DNA-labeled APEX2 or WGA-HRP was compared using 5e5 cells. In order to eliminate the possibility of suspension cell-specific results, we used a popular cell line model of pancreatic cancer, KP-4. We observed that the WGA-HRP identified slightly more PM annotated proteins in the Uniprot Gene Ontology Cellular Component Plasma Membrane (GOCC-PM) database (>2 unique peptides, found in all replicates) relative to DNA-APEX2, totaling 501 and 467, respectively. Notably, the number of IDs for both cell-tethered enzymes was higher than their untethered counterparts, with HRP identifying 389 cell surface proteins and APEX2 identifying 247 (*Figure 2C*). Importantly, in the upset plot shown, the group with the highest intersection includes all four enzyme contexts, showcasing the reproducibility of labeling through a similar free-radical-based mechanism. The cell-tethered biotinylators also showed heightened surface enrichment compared to their untethered counterparts, as illustrated by the higher overall intensities for proteins annotated to the PM (*Figure 2—figure supplement 2A and B*). As equal amounts of total protein are injected on the LC-MS/MS instrument, the higher intensities for PM proteins suggest that localizing the enzyme to the membrane increases labeling of the membrane compartment, which we have previously observed with other enzymatic reactions (*Weeks et al., 2021*).

As the mode of tethering WGA-HRP involves GlcNAc and sialic acid glycans, we wanted to determine whether there was a bias toward Uniprot annotated 'Glycoprotein' versus 'Non-Glycoprotein' surface proteins identified across the WGA-HRP, APEX2-DNA, APEX2, and HRP labeling methods. We looked specifically at surface annotated proteins found in the SURFY database, which is the most stringent surface protein database and requires that proteins have a predicted transmembrane domain (*Bausch-Fluck et al., 2018*). We performed this analysis by measuring the average MS1 intensity across the top three peptides (label-free quantification [LFQ] area) for SURFY glycoproteins and non-glycoproteins for each sample and dividing that by the total LFQ area found across all GOCC-PM annotated proteins detected in each sample. We found similar normalized areas of non-glycosylated surface proteins across all samples (*Figure 2—figure supplement 3*). If a bias existed toward glycosylated proteins in the WGA-HRP compared to the glycan agnostic APEX2-DNA sample, then we would have seen a larger percentage of non-glycosylated surface proteins identified in APEX2-DNA over WGA-HRP. Due to the large labeling radius of the HRP enzyme, we find it unsurprising that the WGA-HRP method is able to capture non-glycosylated proteins on the surface to the same degree (*Rees et al., 2015*). There is a slight increase in the area percentage of glycoproteins detected in the WGA-HRP compared to the APEX2-DNA sample, but this is likely due to the fact that a greater number of surface proteins in general are detected with WGA-HRP. As HRP is known to have faster kinetics compared to APEX2, it was anticipated that WGA-HRP would outperform DNA-APEX2 in cell surface protein identifications. The heightened labeling of WGA-HRP was consistent with every cell line tested, including another pancreatic cancer model, PaTu8902, which resulted in more proteins identified in the WGA-HRP sample over DNA-APEX2 for both 1 and 2 min time points (*Figure 2—figure supplement 4*).

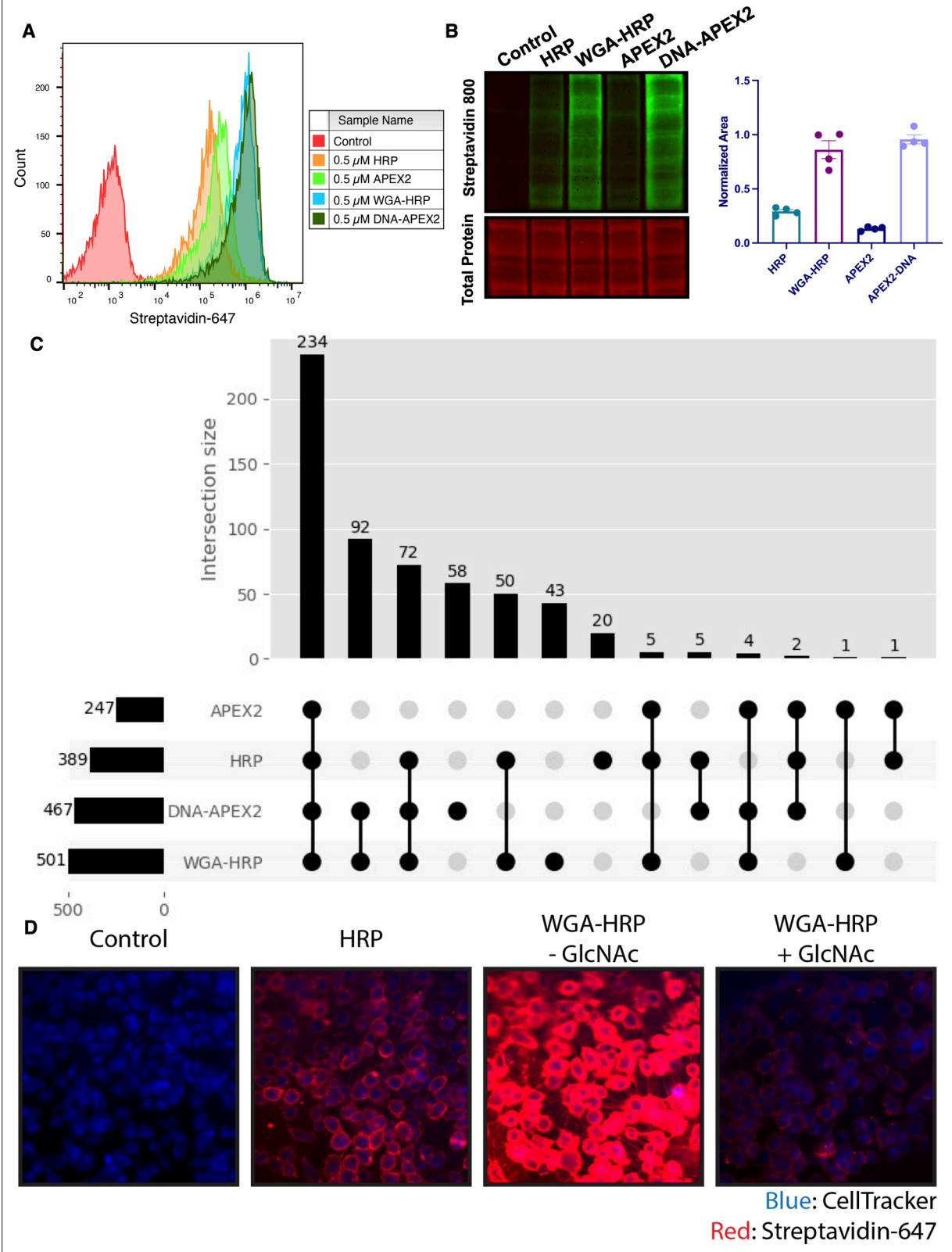

**Figure 2.** Membrane-localized peroxidases increases membrane proteome biotinylation compared to nontethered counterparts. (**A**) Biotinylation of Expi293 cells treated with free enzyme (APEX2 or HRP) or cell-tethered enzyme (DNA-APEX2 or WGA-HRP) shown by flow cytometry. Signal is read out using Streptavidin-AlexaFluor-647. (**B**) Comparison of cell labeling with either free enzyme (APEX2 or HRP) or cell-tethered enzyme (DNA-APEX2 or WGA-HRP) shown by Streptavidin-800 western blot and total protein stain. Normalized area is plotted to the right. (**C**) Number of cell membrane

*Figure 2 continued on next page*

*Figure 2 continued*

proteins identified by mass spectrometry (>2 unique peptides, <1% FDR, found in both biological replicates) after treating 500,000 KP-4 pancreatic cancer cells with either free enzyme (APEX2 or HRP) or cell-tethered enzyme (DNA-APEX2 or WGA-HRP). (**D**) Microscopy images depicting extent of labeling with free HRP compared to WGA-HRP with and without the blocking sugar GlcNAc. All western blot images, microscopy images, mass spectrometry data, and flow cytometry data are representative of two biological replicates.

The online version of this article includes the following source data and figure supplement(s) for figure 2:

**Figure supplement 1.** Optimization of APEX2 concentrations on cell by flow cytometry.

**Figure supplement 2.** Rank ordered intensities for surface annotated proteins detected in tethered and untethered enzyme samples.

**Figure supplement 3.** Comparison of enrichment for glycosylated and non-glycosylated proteins.

**Figure supplement 4.** Total plasma membrane (PM) protein identifications for DNA-APEX2 and WGA-HRP labeling experiments as a function of time.

**Figure supplement 4—source data 1.** Mass spectrometry analysis results table.

**Figure supplement 5.** WGA-HRP labeling is N-acetyl-D-glucosamine (GlcNAc) dependent.

**Figure supplement 6.** WGA-HRP can be used to label adherent cells on-plate.

**Source data 1.** Uncropped western blots.

**Source data 2.** Mass spectrometry analysis results table.

To confirm that the improved labeling by WGA-HRP was due to the binding of sugar units on the cell surface, we performed a sugar-blocking experiment with WGA-HRP using N-acetyl-D-glucosamine (GlcNAc) that would block the conjugate from binding to the cell. By preincubating WGA-HRP with excess GlcNAc, the ability of WGA-HRP to label the cell surface was markedly lower than WGA-HRP without GlcNAc as observed by microscopy (*Figure 2D*). A similar effect was also seen by flow cytometry (*Figure 2—figure supplement 5*). In addition, we also tested an on-plate protocol for simpler cell surface labeling of adherent KP-4 cells. We showed that cell surface labeling in this manner was comparable to labeling when the cells were in suspension (*Figure 2—figure supplement 6*).

As WGA-HRP consistently outperformed DNA-APEX2 by proteomics and represents a more facile method amenable to broad application in the field, we chose to compare the proteomic labeling results of WGA-HRP to other standard cell surface labeling methods (sulfo-NHS-LC-LC-biotin and biocytin hydrazide) on a prostate epithelial cell line, RWPE-1 with and without oncogenic c-Myc over-expression. Sulfo-NHS-LC-LC-biotin reacts with primary amines to form amide conjugates, but has notoriously high background contamination with intracellular proteins (*Weekes et al., 2010*). Biocytin hydrazide labeling is a two-step process that first involves oxidizing vicinal diols on glycoproteins at the cell surface, then reacting the aldehyde byproducts with biocytin hydrazide (*Elschenbroich et al., 2010*). Both WGA-HRP and biocytin hydrazide had similar levels of cell surface enrichment on the peptide and protein level when cross-referenced with the SURFY curated database for extracellular surface proteins with a predicted transmembrane domain (*Figure 3—figure supplement 1A*). Sulfo-NHS-LC-LC-biotin and whole-cell lysis returned the lowest percentage of cell surface enrichment, suggesting that a larger portion of the total sulfo-NHS-LC-LC-biotin protein identifications were of intracellular origin, despite the use of the cell-impermeable format. These same enrichment levels were seen when the datasets were searched with the curated GOCC-PM database, as well as Uniprot's entire human proteome database (*Figure 3—figure supplement 1B*). Of the proteins quantified across all four conditions, biocytin hydrazide and WGA-HRP returned higher overall intensity values for SURFY-specified proteins than either sulfo-NHS-LC-LC-biotin or whole-cell lysis. Importantly, although biocytin hydrazide shows slightly higher cell surface enrichment compared to WGA-HRP, we were unable to perform the comparative analysis at 500,000 cells—instead requiring 1.5 million—as the biocytin hydrazide protocol yielded too few cells for analysis. All three methods were highly reproducible across replicates (*Figure 3—figure supplement 2A-C*). Compared to existing methods, WGA-HRP not only labels cells efficiently with much lower input material requirements, but is also able to enrich cell surface proteins to a similar extent in a fraction of the time.

## WGA-HRP identifies surface markers of Myc-driven prostate cancer in both cells and small EVs

Prostate cancer remains one of the most common epithelial cancers in the elderly male population, especially in Western nations (*Litwin and Tan, 2017*; *Rawla, 2019*). While metastatic progression

of prostate cancer has been linked to many somatic mutations and epigenetic alterations (PTEN, p53, Myc, etc.), more recent work determined that alterations in Myc occur in some of the earliest phases of disease, that is, in tumor-initiating cells (*Koh et al., 2010*). This finding promotes the idea that the development of early stage diagnostic tools that measure these Myc-driven disease manifestations could improve detection and overall patient disease outcomes (*Koh et al., 2010*; *Rebello et al., 2017*). One mode of early detection that has gained prominence is the use of prostate cancer-derived exosomes in patient serum and urine (*Duijvesz et al., 2013*; *McKiernan et al., 2016*). Small EVs are known to play important roles in the progression of prostate cancer, including increasing tumor progression, angiogenesis, metastasis, and immune evasion, making this subcellular particle an extremely informative prognostic tool for disease progression (*Akoto and Saini, 2021*; *Lorenc et al., 2020*; *Saber et al., 2020*).

To elucidate promising targets in Myc-induced prostate cancer, we utilized our WGA-HRP method to biotinylate cells from both normal epithelial prostate cells (RWPE-1 Control) and oncogenic Myc-induced prostate cancer cells (RWPE-1 Myc, *Figure 3A*). Importantly, by using an isogenic system, we are able to delineate specific Myc-driven protein expression changes, which could be helpful in the identification of noninvasive, early detection diagnostics for cancer driven by early Myc induction. In addition to having marked overexpression of c-Myc in the RWPE-1 Myc cells compared to Control, they also grow with a more mesenchymal and elongated morphology compared to their Control cell counterparts (*Figure 3B*), which would suggest large cell surface changes upon oncogenic Myc induction. We initially used WGA-HRP to quantitatively compare the cell surface profiles of Myc-induced prostate cancer to Control cells and found large and bidirectional variations in their surfaceomes (*Figure 3C and D*). We have highlighted the 15 most upregulated proteins in each cell type that are annotated as extracellular surface proteins in the GOCC-PM database. Proteins that are also found in the most restrictive SURFY database that requires a predicted transmembrane domain are bolded in the figure. Proteins annotated to be secreted (Uniprot) from the cell are italicized (*Bausch-Fluck et al., 2018*). All significantly (p<0.05) upregulated SURFY and secreted proteins (>2-fold) are listed with corresponding fold-change quantification (*Figure 3—source data 3*). Vimentin, a marker known to be associated with epithelial-to-mesenchymal transition (EMT), showed heightened expression, in the context of oncogenic Myc, as well as CDH2 (N-Cadherin), another marker of EMT (*Figure 3D*; *Liu et al., 2015*; *Nakajima et al., 2004*). While vimentin has traditionally been described as an intracellular protein, an extracellular membrane-bound form has been found to be important in the context of cancer (*Mitra et al., 2015*; *Noh et al., 2016*). ANPEP and fibronectin-1 were also highly upregulated. Notably, a number of HLA molecules were downregulated in the Myc-induced RWPE cells, consistent with prior findings of loss of MHC class I presentation in prostate cancer (*Blades et al., 1995*; *Cornel et al., 2020*; *Dhatchinamoorthy et al., 2021*). A subset of these findings was verified by both western blot (*Figure 3E*) and microscopy (*Figure 3F*), which highlights the robustness of the protein quantification afforded by using this method.

Next, we wanted to use our WGA-HRP method to quantify cell surface proteins on a sucrose-gradient purified population of small EVs derived from both normal epithelial prostate cells (RWPE-1 Control) and oncogenic Myc-induced prostate cancer cells (RWPE-1 Myc, *Figure 4A*). While sucrose gradient centrifugation generally yields a mixture of vesicle populations, we wanted to confirm that our preparation enriched for vesicles originating from multiple vesicular bodies (MVBs), consistent with an exosome-enriched sample (*Mathieu et al., 2021*). To do so, we prepared EVs from both Control and Myc cells and carried out LFQ mass spectrometry on the whole EV lysates. Following normalization for cell number, we found the Myc cells produced nearly 40% more EVs than the corresponding control cells, which is consistent with previous work that has shown Myc overexpression yields higher quantities of EVs (*Kilinc et al., 2021*). After averaging the intensities between Control and Myc derived EVs, many of the highest intensity proteins (CD9, SDCB1, CD81, LAMP1, LAMP2, ALIX, and CD63) are consistent with MVB-derived vesicle biogenesis, supporting that the sample was likely enriched in EVs rather than other sedimentable particles that can co-isolate during centrifugation (*Figure 4B*). Due to the complex process and extensive washing involved in small EV isolation, many standard labeling methods are not amenable for EV surface labeling. Using WGA-HRP, we are able to biotinylate the small EVs before the sucrose gradient purification and isolation steps (*Figure 4C*). This delineated an important subset of proteins that are differentially expressed under Myc induction, which could serve as interesting targets for early detection in patient urine or serum (*Figure 4D*). All significantly

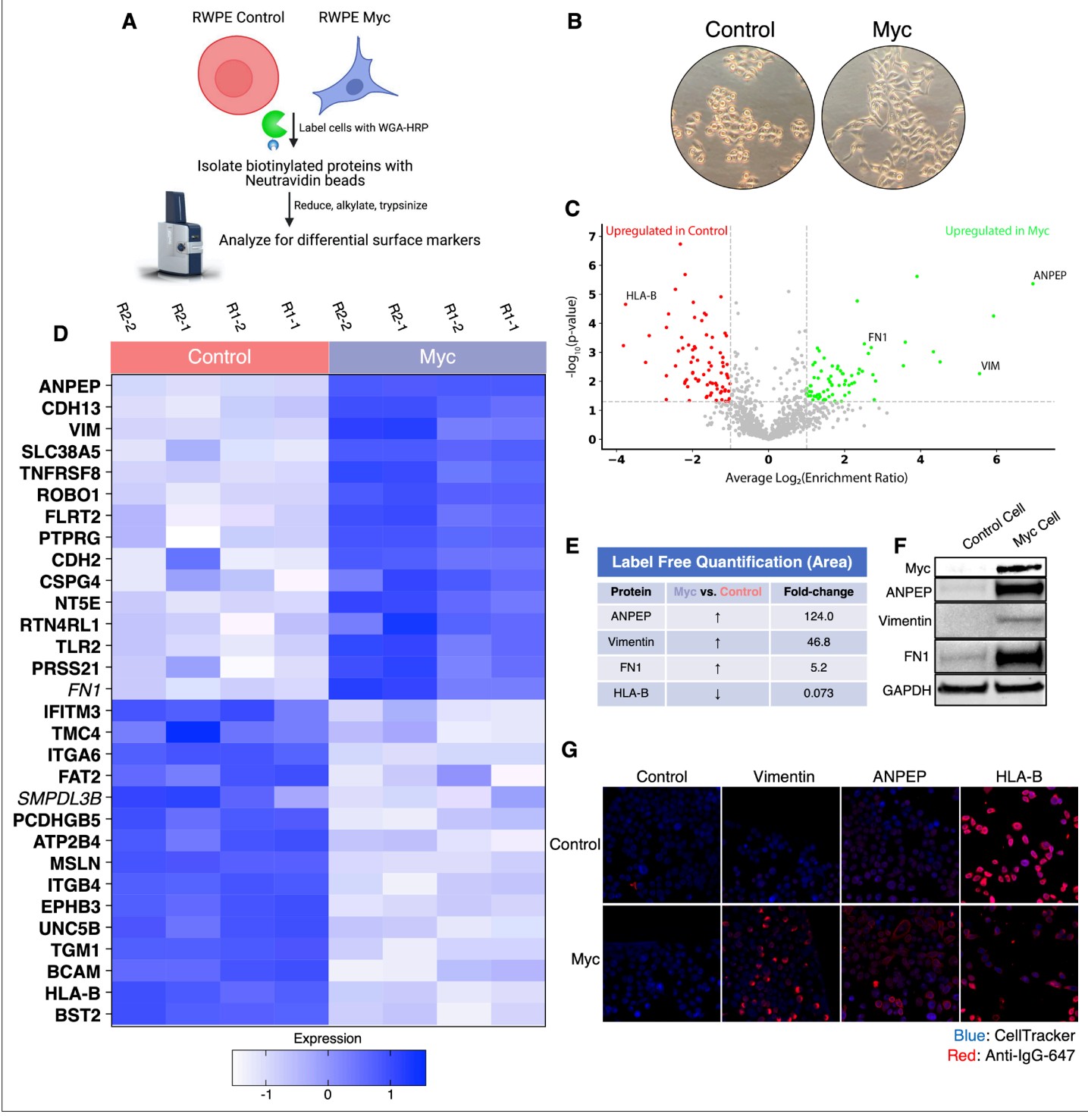

**Figure 3.** WGA-HRP identifies a number of enriched markers on Myc-driven prostate cancer cells. (**A**) Overall scheme for biotin labeling, and label-free quantification (LFQ) by LC-MS/MS for RWPE-1 Control and Myc overexpression cells. (**B**) Microscopy image depicting morphological differences between RWPE-1 Control and RWPE-1 Myc cells after 3 days in culture. (**C**) Volcano plot depicting the LFQ comparison of RWPE-1 Control and Myc labeled cells. Red labels indicate upregulated proteins in the RWPE-1 Control cells over Myc cells and green labels indicate upregulated proteins in the RWPE-1 Myc cells over Control cells. All colored proteins are at least two-fold enriched in either dataset between four replicates (two technical, two biological, p<0.05). (**D**) Heatmap of the 30 most upregulated transmembrane (bold) or secreted (italics) proteins in either RWPE-1 Control or Myc cells. Scale indicates intensity, defined as (LFQ Area−Mean LFQ Area)/Standard Deviation. (**E**) Table indicating fold-change of most differentially regulated proteins by LC-MS/MS for RWPE-1 Control and Myc cells. (**F**) Upregulated proteins in RWPE-1 Myc cells (Myc, ANPEP, Vimentin, and FN1) are confirmed by western blot. (**G**) Upregulated surface proteins in RWPE-1 Myc cells (Vimentin, ANPEP, and FN1) are detected by immunofluorescence

*Figure 3 continued on next page*

*Figure 3 continued*

microscopy. The downregulated protein HLA-B by Myc overexpression was also detected by immunofluorescence microscopy. All western blot images and microscopy images are representative of two biological replicates. Mass spectrometry data is based on two biological and two technical replicates (N=4).

The online version of this article includes the following source data and figure supplement(s) for figure 3:

**Source data 1.** Uncropped western blots.

**Source data 2.** Mass spectrometry analysis results table.

**Source data 3.** List of proteins comparing enriched targets (>2-fold) in Myc cells versus Control cells.

**Figure supplement 1.** Comparison of surface enrichment between replicates for different mass spectrometry methods.

**Figure supplement 1—source data 1.** Mass spectrometry analysis results.

**Figure supplement 2.** Comparison of replicates for different mass spectrometry methods shows that WGA-HRP has comparable reproducibility to NHS-Biotin and Hydrazide labeling.

upregulated SURFY and secreted proteins (>1.5-fold) are listed with corresponding fold-change quantification (*Figure 4—source data 4*). This subset included fibronectin-1 (FN1) and ANPEP (*Figure 4E*), which were further validated by quantitative western blotting (*Figure 4F*). A subset of these targets displays similar phenotypic changes to the parent cell, suggesting that they could be biomarker candidates for noninvasive indicators of disease progression. While certain proteins are shuttled to vesicle compartments largely based on the extent of expression in the parent cell (Control: IFITM3, BST2, and HLA-B, Myc: ANPEP, SLC38A5, and FN1), remarkably some proteins are singled out for small EV packaging, indicating a pronounced differential shuttling mechanism of the proteome between cells and EVs. This pattern was recapitulated in both the RWPE-1 Control cells and corresponding EVs, as well as the Myc cells and EVs, where the majority of markers were unique to either cellular or EV origin (*Figure 4—figure supplement 1*). These protein targets are of extreme interest for not only biomarker discovery, but also understanding the role of small EVs in secondary disease roles, such as interfering with immune function or priming the metastatic niche (*Costa-Silva et al., 2015*).

Due to the difficulty of proteomic characterization of vesicular populations, our current understanding of EV protein shuttling remains limited. Prior proteomic EV analysis has involved whole EV preparations, which lacks a surface protein enrichment step (*Bandu et al., 2019*; *Bilen et al., 2017*; *Hosseini-Beheshti et al., 2012*). Not only is whole EV lysate analysis less advantageous for the specific identification of cell surface proteins on EVs, but it makes it impossible to compare cellular and EV samples due to the inherent surface area-to-volume differences between cells and the vesicles they produce (*Doyle and Wang, 2019*; *Santucci et al., 2019*). Our WGA-HRP method allows us to compare surface proteins between small EV populations, as well as between small EV and cell samples (*Figure 5A*). By principle component analysis (PCA), each sample separates by oncogenic status and origin (*Figure 5B*). Indeed, when performing functional annotation for each gene cluster defined by the PCA, 'extracellular exosome' and 'extracellular vesicle' are the highest ranking annotation features differentiating the EVs from their parent cells (*Figure 5C*). Through this comparison, we were able to delineate a host of proteins that were upregulated in EVs over their parent cells and vice versa (*Figure 5D*). Notably, secreted proteins were more highly represented in the EV surface proteome compared to cells. A subset of proteins was highly upregulated in the small EVs compared to parent cell, including ITIH4, MFGE8, TF, DSG1, TSPAN14, AZGP1, and IGSF8 (*Figure 5E*), and a subset of the findings was validated by western blot (*Figure 5F*). All significantly (p<0.05) upregulated SURFY and secreted proteins (>2-fold) are listed with corresponding fold-change quantification (*Figure 5—source data 3*). The samples showed good overlap between replicates across all four datasets, with cellular and EV samples clustering by origin and oncogenic status (*Figure 5—figure supplement 1*). To our knowledge, this is the first experiment to wholistically characterize the surface proteome of both small EVs and parental cells. These data strongly suggest that protein triage into EVs is a controlled process, enabling only a subset of the cell surface proteome to be shuttled to this important compartment. Our data shows that there are a variety of pan-prostate-EV markers, notably lactadherin (MFGE8), serotransferrin (TF), inter-alpha-trypsin inhibitor (ITIH4), immunoglobulin superfamily 8 (IGSF8), desmoglein-1 (DSG1), tetraspanin-14 (TSPAN14), and zinc-alpha-2-glycoprotein (AZGP1) (*Figure 5D*), which do not seem to be Myc-specific. Some of the pan-prostate EV targets in our data have previously been linked to cancer-specific contexts, and we show here that they are also

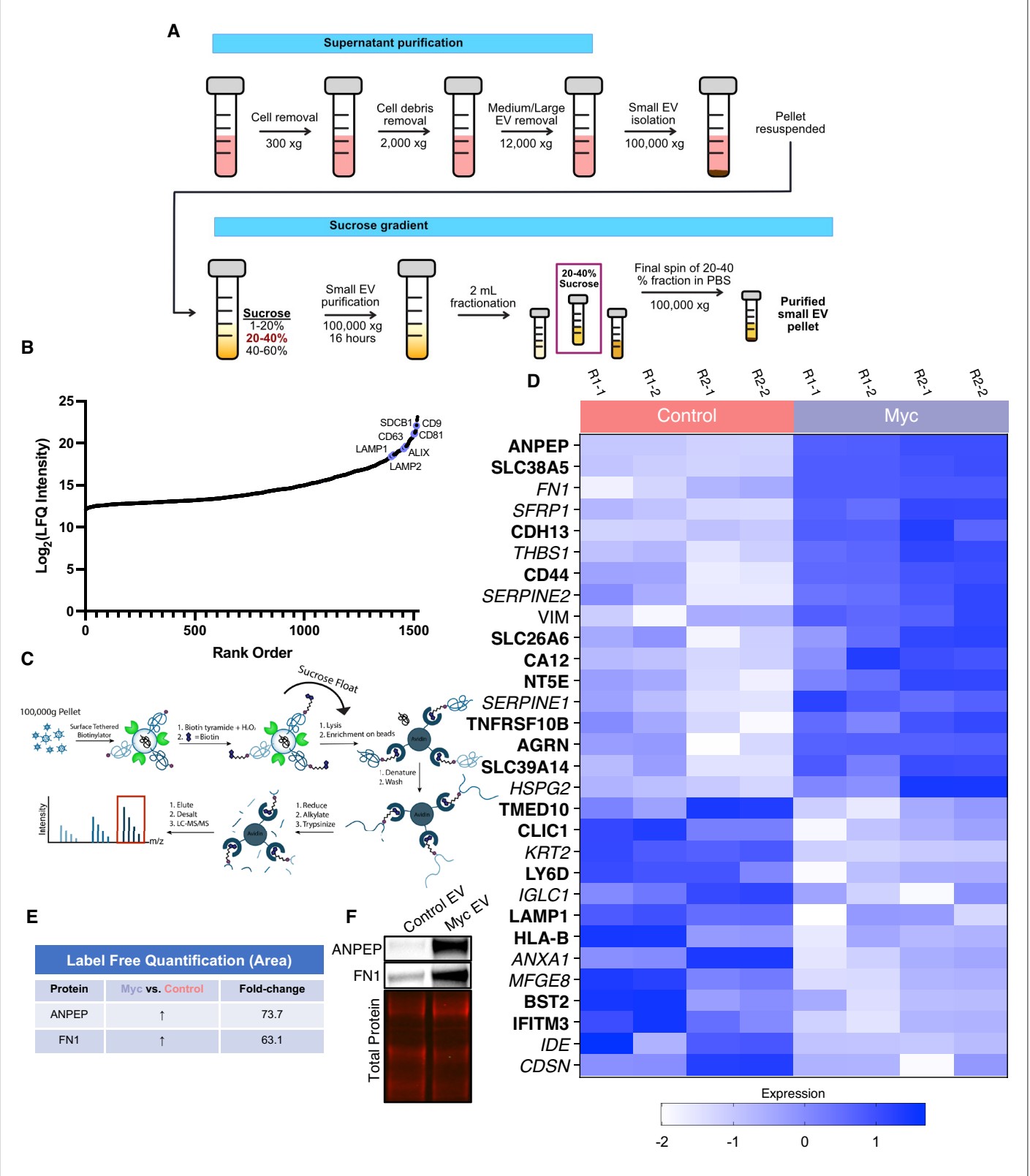

**Figure 4.** WGA-HRP identifies a number of enriched markers on Myc-driven prostate cancer EVs. (**A**) Workflow for small EV isolation from cultured cells. (**B**) Labeled proteins indicating canonical exosome markers (ExoCarta Top 100 List) detected after performing label-free quantification (LFQ) from whole EV lysate. The LFQ intensities were averaged for Control and Myc EVs, and the resulting protein list is graphed from least abundant to most abundant. (**C**) Workflow of EV labeling and preparation for mass spectrometry. (**D**) Heatmap of the 30 most upregulated proteins in either RWPE-1 Control or

*Figure 4 continued on next page*

*Figure 4 continued*

Myc EVs. Scale indicates intensity, defined as (LFQ Area−Mean LFQ Area)/Standard Deviation. (**E**) Table indicating fold-change of most differentially regulated proteins by LC-MS/MS for RWPE-1 Control and Myc cells. (**F**) Upregulated proteins in RWPE-1 Myc EVs (ANPEP and FN1) are confirmed by western blot. Mass spectrometry data is based on two biological and two technical replicates (N=4). Due to limited sample yield, one replicate was performed for the EV western blot. EV, extracellular vesicle.

The online version of this article includes the following source data and figure supplement(s) for figure 4:

**Figure supplement 1.** Venn diagram comparing enriched targets (>2-fold) in cells and EVs.

**Source data 1.** Uncropped western blots.

**Source data 2.** Whole EV mass spectrometry analysis results table.

**Source data 3.** Mass spectrometry analysis results table.

**Source data 4.** List of proteins comparing enriched targets (>1.5-fold) in Myc EVs versus Control EVs.

found on Control EVs (*Shimagaki et al., 2019*; *Tutanov et al., 2020*; *Philley et al., 2017*). Our work suggests that these markers are more broadly associated with small EVs, regardless of disease status, outlining an expanded set of targets to probe these vital compartments.

## Discussion

The importance of understanding and characterizing cellular and EV membrane compartments is vital for improving our understanding of vesicle biogenesis. New, improved methodologies amenable to small-scale and rapid surface proteome characterization are essential for continued development in the areas of therapeutics, diagnostics, and basic research. We sought to develop a simple, rapid surface protein labeling approach that was compatible with small sample sizes, while remaining specific to the cell surface. We took advantage of fast peroxidase enzymes and either complementary lipidated DNA technology (DNA-APEX2) or the glycan-binding moiety wheat germ agglutinin (WGA-HRP) and demonstrated that tethering was much more effective than soluble addition, with increases in protein identification of between 30% and 90%. Additionally, we compared WGA-HRP to the existing methods, sulfo-NHS-LC-LC-biotin and biocytin hydrazide. While these alternative methods are robust, they are unable to capture time-sensitive changes and are either plagued by low selectivity/specificity (NHS-Biotin) (*Weekes et al., 2010*) or the requirement for large sample inputs (biocytin hydrazide).

There are many advantages of our new methods over the current cell surface labeling technologies. Compared to both sulfo-NHS-LC-LC-biotin and biocytin hydrazide, WGA-HRP experiments require 2 min instead of 30 or 120 min, respectively. It is also able to enrich cell surface proteins much more efficiently than sulfo-NHS-LC-LC-biotin labeling. Furthermore, NHS peptide isolation and preparation is complicated due to the reactivity of NHS chemistry toward free-amines, which blocks tryptic and LysC cleavages typically used in proteomics (*Chandler and Costello, 2016*; *Hacker et al., 2017*).

The hydrazide method is highly effective for enriching cell surface proteins, but it is challenging for small sample sizes, due to the two-step labeling process and cell loss from the oxidation step and extensive washing. Additionally, neither NHS-biotin nor biocytin hydrazide are able to capture short time points to encompass dynamic changes at the cell surface. Due to the fast kinetics of peroxidase enzymes (1–2 min), our approaches could enable kinetic experiments to capture rapid post-translational trafficking of surfaces proteins, such as response to insulin, certain drug treatments, T-cell activation and synapse formation, and GPCR activation (*Valitutti et al., 2010*; *Gupte et al., 2019*; *Li et al., 2021*). Another disadvantage of the hydrazide method is that it can only enrich for proteins that are glycosylated at the cell surface, and it is estimated that 10%–15% of cell surface proteins are not glycosylated (*Apweiler et al., 1999*). Glycosylation patterns also readily change during tumorigenesis, which can alter the quantification of glycan-based labeling methods, such as biocytin hydrazide (*Reily et al., 2019*). While the WGA-HRP method requires glycosylated proteins to be present to bind, it is still able to label non-glycosylated proteins nearby due to its large labeling radius. It is a possibility that certain cells may have low or uneven levels of glycosylation on their surfaces. In these cases, the DNA-APEX2 method can be utilized to obtain effective labeling. However, both these peroxidase-based methods require the presence of tyrosine residues (natural abundance 3.3%) to react with the biotin-tyramide radical, which is not equally abundant in all proteins (*Dyer, 1971*).

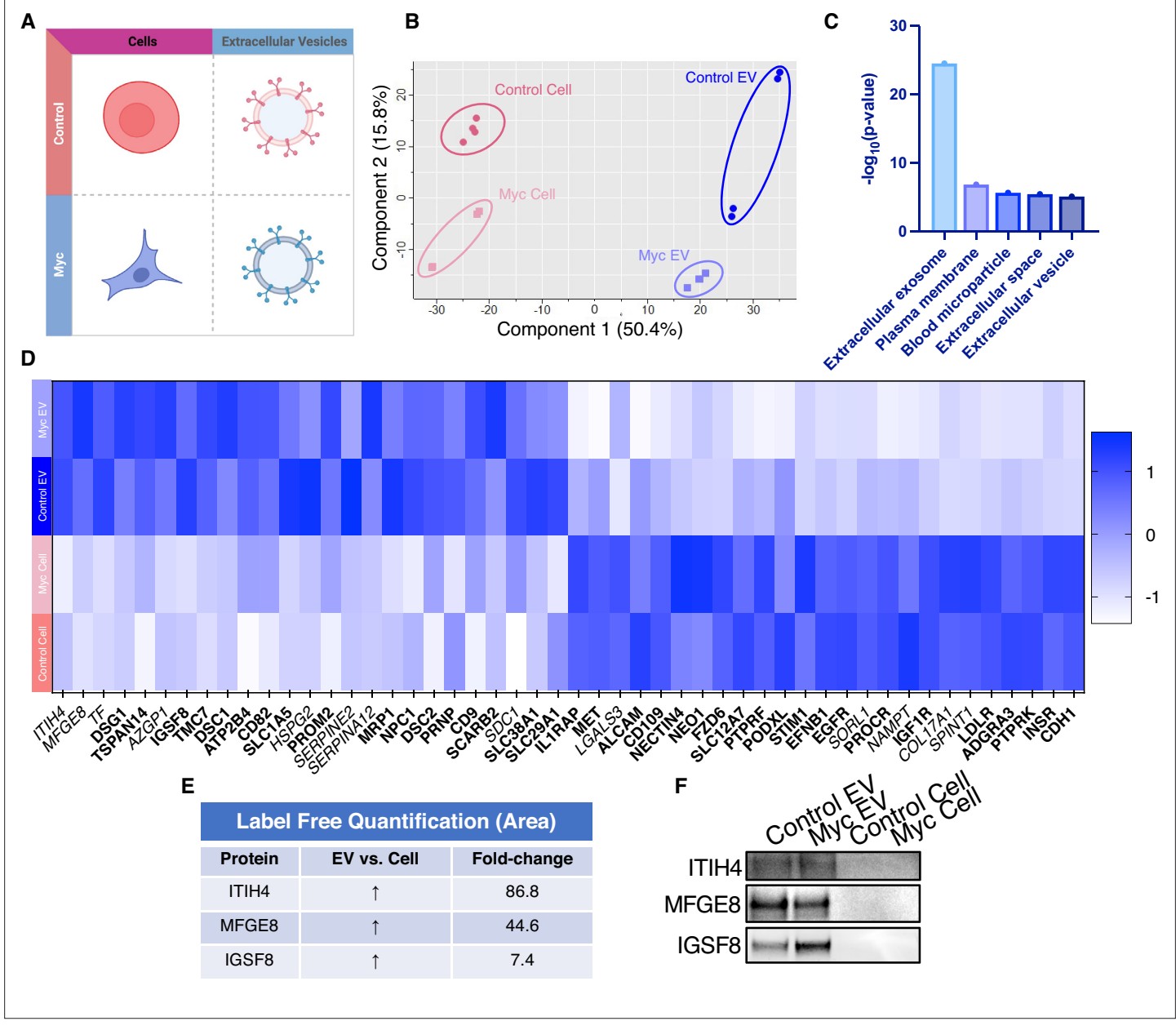

**Figure 5.** WGA-HRP identifies a number of EV-specific markers that are present regardless of oncogene status. (**A**) Matrix depicting samples analyzed during LFQ comparison–Control and Myc cells, as well as Control and Myc EVs. (**B**) Principle component analysis (PCA) of all four groups analyzed by LFQ. Component 1 (50.4%) and component 2 (15.8%) are graphed. (**C**) Functional annotation was performed for each gene cluster using DAVID Bioinformatics Resource 6.8 and the highest ranking annotation features for the EV-specific gene cluster are shown. (**D**) Heatmap of the 50 most upregulated proteins in either RWPE-1 cells or EVs. Proteins are listed in decreasing order of expression with the most highly expressed proteins in EVs on the far left and the most highly expressed proteins in cells on the far right. Averages from all four replicates of each sample type are graphed. Scale indicates intensity, defined as (LFQ Area−Mean LFQ Area)/Standard Deviation. Extracellular proteins with annotated transmembrane domains are bolded and annotated secreted proteins are italicized. (**E**) Table indicating fold-change of most differentially regulated proteins by LC-MS/MS for RWPE-1 EVs compared to parent cells. (**F**) Western blot showing the EV-specific marker ITIH4, IGSF8, and MFGE8. Mass spectrometry data is based on two biological and two technical replicates (N=4). Due to limited sample yield, one replicate was performed for the EV western blot. EV, extracellular vesicle; LFQ, label-free quantification.

The online version of this article includes the following source data and figure supplement(s) for figure 5:

**Source data 1.** Uncropped western blots.

**Source data 2.** Mass spectrometry analysis results table.

**Source data 3.** List of proteins comparing enriched targets (>2-fold) in Control EVs versus Control cells and Myc EVs versus Myc cells.

**Figure supplement 1.** Heatmap comparison of biological and technical replicates of RWPE-1 Control/Myc cells and EVs.

With the WGA-HRP method, we were able to compare the surfaceome of small EVs to parental cells for Myc-induced prostate cancer cells and identified proteins that were upregulated in Myc-induced cells and EVs, as well as proteins that were differentially shuttled between EVs and parental cells. We found a number of Myc-specific markers in our study, which were enriched in both Myc EV and Myc Cell samples. These include ANPEP, SLC38A5, FN1, CDH13, VIM, and CA12. ANPEP is a membrane-bound ectopeptidase that degrades N-termini with neutral amino acids and is found 140-fold upregulated in the Myc-induced cell compared to the Control cell and 49-fold upregulated in the Myc-induced EV compared to Control EV. This peptidase has been associated with angiogenesis and cancer growth (*Guzman-Rojas et al., 2012*; *Sorensen et al., 2013*; *Wickström et al., 2011*). Recent studies have shown ANPEP is systematically upregulated on isogenic cell lines expressing proliferative oncogenes *Leung et al., 2020*; *Martinko et al., 2018* or in tubular sclerosis bladder cancers *Wei et al., 2020*, suggesting it is commonly upregulated in cancers. The second most differentially expressed protein between the Myc and Control samples was SLC38A5 (23- and 73-fold upregulated in cells and EVs, respectively). SLC38A5 is a glutamine co-transporter and has previously been shown to be a downstream target of c-Myc in glutamine-addicted cancers. Moreover, given that SLC38A5-based glutamine transport leads to proton flux and intracellular alkanization, overexpression of SLC38A5 has also been hypothesized to be a strategy for pH regulation in cancer cells that regularly experience intracellular acidification due to high glycolytic flux (*Bhutia and Ganapathy, 2016*; *Wise et al., 2008*). Additionally, Fibronectin-1 (FN1) was also found to be upregulated in Myc samples over Control samples (5- and 63-fold upregulated in cells and EVs, respectively) and has been shown to drive all stages of tumorigenesis (*Wang and Hielscher, 2017*). Importantly, FN1 provides an extracellular scaffold by which other matrix proteins can be deposited. Through these interactions with matrix proteins and cell-associated integrins, FN1 regulates cellular fate decisions, proliferation, and metastasis (*Efthymiou et al., 2020*).

While some proteins were present in both the EV and cellular samples, others were only found enriched in Myc EVs. THBS1, also known as thrombospondin-1, was over ten-fold upregulated in Myc EVs over Control EVs. Interestingly, this relationship was not found in the parent cells, which suggests that THBS1 is differentially shuttled into oncogenic EVs. The role of this protein has newly been associated with the growth and metastasis of glioblastoma and a potential serum prognostic factor in myeloid leukemia (*Zhu et al., 2019*; *Daubon et al., 2019*). Moreover, using a model of THBS1 overexpressing breast cancer, recent work has shown that exosomes laden with THBS1 promote cancer cell migration via disruption of the endothelial barrier (*Cen et al., 2019*).

Another such target is CD44, which was over eight-fold upregulated in the Myc EVs over Control EVs. CD44 has long been known to drive cancer progression and aberrant cell signaling (*Chen et al., 2018*). Recently, CD44 has also been found to be preferentially loaded into cancer-derived exosomes and has been implicated in driving chemoresistance in a model of doxorubicin-treated breast cancer (*Wang et al., 2020*). Similarly, it has been shown that exosome-mediated transfer of CD44 from cells with high metastatic potential promoted migratory behavior in neighboring cells with low metastatic potential (*Shen et al., 2021*). These targets delineate an important subset of proteins that are triaged into EVs and could play long-range roles in promoting tumorigenesis and downstream metastasis (*Costa-Silva et al., 2015*; *Demory Beckler et al., 2013*; *Hoshino et al., 2015*; *Peinado et al., 2012*).

As research shifts into analyzing native biological samples from EVs to xenograft models or patient biopsies, it will become increasingly important to develop sensitive, effective methods to label these small samples sizes. It is our hope that these tools will provide much needed avenues by which to pursue pressing biological questions in the areas of diagnostic and therapeutic development, as well as basic research.

## Materials and methods

### Key resources table

| Reagent type (species) or resource | Designation | Source or reference | Identifiers | Additional information |
|---|---|---|---|---|
| Strain, strain background (*Escherichia coli*) | BL21(DE3)pLysS | Promega | L1195 | – |

*Continued on next page*

| Reagent type (species) or resource | Designation | Source or reference | Identifiers | Additional information |
|---|---|---|---|---|
| Cell line (human) | Expi293F | Thermo Fisher Scientific | A14527 | – |
| Cell line (human) | PaTu8902 | https://doi.org/10.7554/eLife.45313.001 | – | – |
| Cell line (human) | KP-4 | https://doi.org/10.7554/eLife.45313.001 | – | – |
| Cell line (human) | RWPE-1 | https://doi.org/10.1073/pnas.2018861118 | – | Control and Myc overexpression lines |
| Antibody | Anti-HisTag-650 (mouse monoclonal) | Invitrogen | MA1-21315-D650 | 1:100, flow cytometry, immunocytochemistry |
| Antibody | Biotin-conjugated anti-HRP (rabbit polyclonal) | Rockland | 200-4638-0100 | 1:100, flow cytometry, immunocytochemistry |
| Antibody | Anti-ANPEP (sheep polyclonal) | R&D Systems | AF3815 | 1:1000 |
| Antibody | Anti-Vimentin (rabbit monoclonal) | Cell Signaling Technology | 5741S | 1:1000 |
| Antibody | Anti-FN1 (rabbit polyclonal) | Abcam | ab2413 | 1:1,000 |
| Antibody | Anti-HLA-B (rabbit polyclonal) | ProteinTech | 17260-1-AP | 1:1,000 |
| Antibody | Anti-ITIH4 (rabbit polyclonal) | Atlas antibodies | HPA003948 | 1:1,000 |
| Antibody | Anti-MFGE8 (rabbit polyclonal) | Thermo Fisher Scientific | PA5-82036 | 1:1,000 |
| Antibody | Anti-IGSF8 (goat polyclonal) | R&D Systems | AF3117-SP | 1:1,000 |
| Antibody | Goat Anti-Rabbit HRP (goat polyclonal) | Thermo Fisher Scientific | 31460 | 1:10,000 |
| Antibody | Rabbit Anti-Sheep HRP (rabbit polyclonal) | Thermo Fisher Scientific | 31480 | 1:10,000 |
| Recombinant DNA reagent | APEX2 | Twist Biosciences | – | – |
| Sequence-based reagent | DBCO-DNA (Conjugated to APEX2) | IDT | – | – |
| Sequence-based reagent | Lipid DNA (Anchor and Co-Anchor) | https://doi.org/10.1038/s41592-019-0433-8 | – | – |
| Peptide, recombinant protein | HRP, Peroxidase | Thermo Fisher Scientific | 31,490 | – |
| Peptide, recombinant protein | WGA, Peroxidase (WGA-HRP) | Vector Laboratories | PL-1026-2 | – |
| Peptide, recombinant protein | IRDye 800CW Streptavidin | Li-Cor | 926-32230 | 1:10,000 |
| Peptide, recombinant protein | IRDye 680RD Donkey anti-Goat IgG | Li-Cor | 925-68074 | 1:10,000 |
| Commercial assay or kit | BCA assay | Thermo Fisher Scientific | 23228 | – |
| Commercial assay or kit | Preomics iST kit | PreOmics | P.O.00027 | – |
| Commercial assay or kit | Pierce quantitative colorimetric peptide assay | Thermo Fisher Scientific | 23275 | – |
| Commercial assay or kit | Expi293 media | Thermo Fisher Scientific | A1435101 | – |

| Reagent type (species) or resource | Designation | Source or reference | Identifiers | Additional information |
|---|---|---|---|---|
| Commercial assay or kit | Keratinocyte SFM media (1×) | Thermo Fisher Scientific | 17005042 | – |
| Commercial assay or kit | MycoAlert PLUS mycoplasma detection kit | Lonza | LT07-703 | – |
| Chemical compound, drug | Trolox | Fisher Scientific | 501176131 | – |
| Chemical compound, drug | Sodium Ascorbate | Sigma-Aldrich | A4034-100G | – |
| Chemical compound, drug | N-acetyl-D-glucosamine | Sigma-Aldrich | A3286-5G | – |
| Chemical compound, drug | $H_2O_2$ (1×) | Sigma-Aldrich | H1009-100ML | – |
| Chemical compound, drug | Hemin-Cl | Sigma-Aldrich | 51280-1G | – |
| Chemical compound, drug | Maleimide DBCO | Click Chemistry Tools | A108P-10 | – |
| Chemical compound, drug | Biotin Tyramide | Sigma-Aldrich | SML2135-50MG | – |
| Chemical compound, drug | Sodium Pyruvate | Sigma-Aldrich | S8636-100ML | – |
| Chemical compound, drug | Sodium Periodate | Sigma-Aldrich | 311448-5G | – |
| Chemical compound, drug | Biocytin Hydrazide | Biotium | 90060 | – |
| Chemical compound, drug | Aniline | Sigma-Aldrich | 242284 | – |
| Chemical compound, drug | Protease Inhibitor Cocktail | Sigma-Aldrich | P8340-1ML | – |
| Chemical compound, drug | High Capacity Neutravidin Resin | Thermo Fisher Scientific | 29204 | – |
| Chemical compound, drug | Poly-D-lysine | Thermo Fisher Scientific | A3890401 | – |
| Chemical compound, drug | Sulfo NHS LC-LC Biotin | Thermo Fisher Scientific | A35358 | – |
| Software, algorithm | FlowJo | FlowJo (https://www.flowjo.com) | RRID:SCR_008520 | – |
| Software, algorithm | GraphPad Prism | Prism (https://www.graphpad.com/scientific-software/prism/) | RRID:SCR_002798 | – |
| Software, algorithm | Morpheus | Morpheus (https://software.broadinstitute.org/morpheus/) | – | – |
| Software, algorithm | FIJI | FIJI (https://fiji.sc) | RRID:SCR_002285 | – |
| Other | Penicillin/streptomycin | Thermo Fisher Scientific | 15-140-122 | – |
| Other | Fetal bovine serum | Gemini Bio-Products | 100-106 | – |
| Other | Trypsin (0.05%) | Life Technologies | 25300054 | – |

## Large-scale APEX2 expression, purification, and heme reconstitution

APEX2 was expressed using previous methods in BL21(DE3)pLysS cells (*Howarth and Ting, 2008*). Briefly, APEX2 expression plasmid was transfected into competent BL21(DE3)pLysS cells and heat shocked for 45 s before being placed on ice. Cells were plated on LB/Carb plates and grown overnight

at 37°C. A single colony was isolated and grown in a mixture of 30 ml of 2XYT+Carb overnight at 37°C while shaking. The overnight culture was combined with 3 L of 2XYT with Carb and placed in a 37°C shaking incubator. At an $OD_{600}$ of 0.6, 100 µg/ml of IPTG was added and the temperature of the incubator was lowered to 30°C. Cells were allowed to incubate for 3.5 hr and were spun down at 6000$g$ for 20 min. Cell pellet was resuspended in protease inhibitor-containing resuspension buffer (5 mM Imidazole, 300 mM NaCl, 20 mM Tris, pH=8) and mixed thoroughly. The mixture was sonicated at 50% (5 s on:15 s off) for 5 min on ice to avoid bubble formation. Lysate was mixed by inversion at 4°C for 15 min and spun down at 19,000$g$ for 20 min. The slurry was introduced to 5 ml of washed Nickel resin slurry and allowed to bind by gravity filtration. The beads were washed 3× with wash buffer (30 mM Imidazole, 300 mM NaCl, 20 mM Tris pH=8) and eluted in 5 ml of elution buffer (250 mM Imidazole, 300 mM NaCl, 20 mM Tris, pH=8) before undergoing buffer exchange into phosphate-buffered saline (PBS).

Enzyme underwent heme reconstitution as per previous methods (*Cheek et al., 1999*). Briefly, 50 mg of hemin-Cl (Sigma-Aldrich) was diluted in 2.0 ml of 10 mM NaOH. The mixture was thoroughly resuspended, then diluted further using 8.0 ml of 20 mM $KPO_4$, pH 7.0, and vortexed extensively. Mixture was spun down at 4000$g$ 2× to eliminate insoluble hemin. APEX2 was diluted at 1:2 in 20 mM $KPO_4$. About 6 ml of heme stock was added to 2 ml of APEX over 20 min and allowed to rotate at 4°C wrapped in tin foil for 3 hr. The mixture was introduced to a column with 20 ml of DEAE Sepharose pre-equilibrated in 20 mM $KPO_4$, pH 7.0 buffer. Enzyme was eluted using 100 mM $KPO_4$ and spin concentrated. To verify complete reconstitution, absorbance was measured at 403 and 280 nm. A403/280>2.0 is considered sufficient for reconstitution. The isolated protein was flash-frozen and stored at –80°C for long-term storage. Each batch of enzyme was run out on a 4%–12% Bis-Tris gel to confirm purity (*Figure 1—figure supplement 1*).

## APEX2 DNA labeling protocol

APEX2 was incubated at 50 µM with 40 molar equivalents of maleimide-DBCO for 5 hr at room temperature (RT) in PBS. The reaction was desalted with Zeba columns (7 kDa cutoff). About 2.5 M equivalents of Azido-DNA was added to the reaction and incubated at 4°C overnight. Successful conjugation was monitored by LC-MS before the mixture was purified by nickel column.

## Cell culture

Prior to all experiments, cells were tested for the presence of mycoplasma (MycoAlert PLUS, Lonza, LT07-703). Expi293 suspension cells were maintained in Expi293 media (Thermo Fisher Scientific, A1435101) while rotating at 125 rpm in a 37°C incubator with 8% $CO_2$. Cells were split every 3 days by diluting into new media. Adherent PaTu8902 and KP-4 cells were grown in pre-warmed Iscove's modified Dulbecco's media (IMDM) supplemented with 10% fetal bovine serum (FBS) (Gemini Bio-Products, 100–106) and 5% Penicillin/Streptomycin (Thermo Fisher Scientific, 15-140-122) at 37°C in a 5% $CO_2$-humidified incubator. Adherent RWPE-1 prostate cells were grown in complete keratinocyte-SFM (Thermo Fisher Scientific; 17005-042) supplemented with bovine pituitary extract (BPE), recombinant EGF, and 5% penicillin/streptomycin at 37°C in a 5% $CO_2$-humidified incubator. The media were exchanged every 2 days. For splitting, cells were lifted with 0.05% Trypsin (Life Technologies) and quenched with 5% FBS before spinning down cells to remove residual trypsin and FBS. Cells were then plated in pre-warmed complete keratinocyte-SFM media.

## Microscopy

Cells were plated at a density of 15,000 cells per well in a 96-well clear bottom plate (Greiner Bio-One, 655090) pre-treated with poly-D-lysine (Thermo Fisher Scientific, A3890401). Cells were allowed 48 hr to reattach and grow undisturbed. Cells were washed 3× in cold PBS. For DNA-APEX2, 100 µl of 0.5 µM enzyme solution was combined with anchor and co-anchor at a final concentration of 1 µm. For all other enzymes, enzyme was combined with PBS at a final concentration of 0.5 µM. For sugar blocking studies, 100 µl of diluted enzyme solution (0.5 µM) was combined with 100 mg/ml N-acetyl-D-glucosamine (Sigma-Aldrich, A3286-5G). Cells were allowed to sit on ice for 5 min to allow WGA to bind fully, as labeling was not altered by increased incubation time (*Figure 1—figure supplement 3*). Biotin tyramide (Sigma-Aldrich, SML2135-50MG) was added to cells with a final concentration of 500 µM before adding 1 mM of $H_2O_2$. Reaction was allowed to continue for 2 min before rinsing cells

3× with 1× quench buffer (10 mM sodium ascorbate+5 mM Trolox+1 mM sodium pyruvate). The cells were rinsed 2× with PBS and crosslinked with 4% PFA for 10 min at RT. Cells were washed 3× with PBS before introduction to 1:100 primary antibody. Primary antibodies used were: HisTag-650 (Invitrogen, MA1-21315-D650), Streptavidin-488 (Thermo Fisher Scientific, S-11223), biotin-conjugated anti-HRP (Rockland, 200-4638-0100), ANPEP (R&D Systems, AF3815), vimentin (Cell Signaling Technology, 5741S), and HLA-B (ProteinTech, 17260-1-AP). Cells were washed 3× in PBS and imaged on an IN Cell Analyzer 6500. Images were processed in Fiji using the Bio-Formats plugin (*Linkert et al., 2010*; *Schindelin et al., 2012*).

## Cell-tethered APEX2, soluble APEX2, cell-tethered WGA-HRP, and soluble HRP cell surface labeling

Cultured cells were grown for 3 days in tissue culture plates and dissociated by addition of versene (PBS+0.05% EDTA). Cells were washed 3× in PBS (pH 6.5), resuspended in PBS (pH 6.5), and aliquoted to 500,000 cells per sample. Samples were resuspended in 100 µl of PBS (pH 6.5). For anchored APEX2 samples, lipidated anchor DNA was allowed to bind for 5 min at 1 µM on ice, followed by 1 µM of lipidated co-anchor DNA on ice for 5 min. About 0.5 µM DNA-labeled APEX2 was allowed to bind on cells for 5 min before final wash with PBS (pH 6.5). For soluble APEX2, WGA-HRP, and soluble HRP samples, cells were resuspended in 0.5 µM of the corresponding enzyme. WGA-HRP was allowed to bind to cells for 5 min on ice. Biotin tyramide was added at a final concentration of 500 µM and mixed thoroughly, before the addition of 1 mM $H_2O_2$. Cells underwent labeling in a heated shaker (500 rpm) at 37°C for 2 min before being quenched with 5 mM Trolox/10 mM Sodium Ascorbate/1 mM Sodium Pyruvate. Cells were washed 2× in quench buffer and spun down. The pellet was either further processed for flow cytometry, western blot, or flash-frozen in liquid nitrogen for mass spectrometry.

## On plate WGA-HRP cell surface labeling

KP-4 cells were grown on a 6-cm tissue culture treated plate and washed 3× with PBS (pH 6.5). About 2 ml of 0.5 µM WGA-HRP in PBS (pH 6.5) was added to the plate, followed by biotin tyramide (0.5 mM final concentration) and $H_2O_2$ (1 mM final concentration). After a 2-min incubation at 37°C, the cells were washed 2× with 5 mM Trolox/10 mM Sodium Ascorbate/1 mM Sodium Pyruvate quenching solution. The cells were washed 1× with PBS before being lifted with versene (PBS+0.05% EDTA). Once lifted, the cells were washed once with PBS and subsequentially processed for flow cytometry analysis.

## Biocytin hydrazide cell surface labeling

Cultured cells were grown for 3 days in tissue culture plates and dissociated by addition of versene (PBS+0.05% EDTA). Cells were washed 3× in PBS (pH 6.5), resuspended in PBS (pH 6.5), and aliquoted to 1.5 million cells per sample. Samples were resuspended in 100 µl of PBS (pH 6.5) and fresh sodium periodate (Sigma-Aldrich, 311448, 1 µl of a 160 mM solution) was added to each sample. The samples were mixed, covered in foil, and incubated while rotating at 4°C for 20 min. Following three washes with PBS (pH 6.5), the samples were resuspended in 100 µl of PBS (pH 6.5) with the addition of 1 µl of aniline (Sigma-Aldrich, 242284, diluted 1:10 in water) and 1 µl of 100 mM biocytin hydrazide (Biotium, 90060). The reaction proceeded while rotating at 4°C for 90 min. The samples were then washed 2× with PBS (pH 6.5) and spun down. The pellet was either further processed for flow cytometry, western blot, or flash-frozen in liquid nitrogen for mass spectrometry.

## Sulfo-NHS-LC-LC-biotin cell surface labeling

Cultured cells were grown for 3 days in tissue culture plates and dissociated by the addition of versene (PBS+0.05% EDTA). Cells were washed 3× in PBS (pH 7.4), resuspended in PBS (pH 8), and aliquoted to 1.5 million cells per sample. Samples were resuspended in 50 µl of PBS (pH 8). An aliquot of EZ-Link Sulfo-NHS-LC-LC-Biotin (Thermo Fisher Scientific, 21338) was resuspended in 150 µl of PBS (pH 8). About 7.5 µl was added to each cell sample and the reaction proceeded while rotating at 4°C for 30 min. The reaction was quenched by the addition of 2.5 µl of 1 M Tris (pH 8.0). The samples were washed 2× in PBS (pH 8.0) and spun down. The pellet was either further processed for flow cytometry, western blot, or flash-frozen in liquid nitrogen for mass spectrometry.

## Flow cytometry for cell surface biotinylation

After labeling and quench washes, the cells were washed once with PBS + 2% BSA to inhibit nonspecific binding. Samples were then incubated with 100 µl Streptavidin-Alexa Fluor 647 (Thermo Fisher Scientific, 1:100 in PBS + 2% BSA). Following a 30-min incubation at 4°C while rocking, samples were washed three times with PBS + 2% BSA. Samples were analyzed in the APC channel and quantified using a CytoFLEX (Beckman Coulter). All flow cytometry data analysis was performed using FlowJo software.

## RWPE-1 small EV isolation and labeling protocol

RWPE-1 Control and Myc cells were plated at 7 million and 4 million cells per plate, respectively, across 16×15 cm$^2$ plates and allowed to grow in normal keratinocyte-SFM media with provided supplements. Small EVs were isolated as previously described (*Poggio et al., 2019*). Briefly, 2 days prior to EV isolation, media was replaced with 15 ml BPE-free keratinocyte-SFM media. For vesicle enrichment, media were isolated after 2 days in BPE-free media and centrifuged at 300$g$ for 10 min at RT, followed by 2000$g$ for 20 min at 4°C. Large debris was cleared by a 12,000$g$ spin for 40 min at 4°C. The precleared supernatant was spun a final time at 100,000$g$ at 4°C for 1 hr to pellet EVs. Isolated EVs were brought up in 50 µl of PBS with 0.5 µM of WGA-HRP and the mixture was allowed to bind on ice for 5 min. WGA-HRP bound vesicles were placed on a shaker (500 rpm) at 37°C before the addition of biotin tyramide (0.5 mM final concentration) and H$_2$O$_2$ (1 mM final concentration). Vesicles underwent labeling for 2 min before being quenched with 5 mM Trolox/10 mM Sodium Ascorbate/1 mM Sodium Pyruvate. Biotinylated small EVs were purified from other sedimentable particles by further centrifugation on a sucrose gradient (20%–60%) for 16 hr at 4°C at 100,000$g$. Precisely, the gradient was loaded using 0%, 20%, 40%, and 60% sucrose fractions from top to bottom. The sample was loaded at the bottom in 60% sucrose and the purified small EVs were isolated in the 20%–40% sucrose fractions. Differential sucrose centrifugation yielded between 3 and 5 µg of small EVs.

## Western blot protocol

Cultured cells were grown in 15-cm$^2$ tissue culture plates and dissociated by addition of versene (PBS + 0.05% EDTA). Cells were washed in PBS (pH 6.5) and resuspended in 100 µl PBS (pH 6.5) at a concentration of 10 million cells/ml in PBS (pH 6.5). Cells were labeled, reaction was quenched with 1× NuPage Loading Buffer, and immediately boiled for 5 min. To enable proper addition of lysate to gel wells, the mixture was thinned with addition of nuclease, and the disulfides were reduced with BME. The samples were subjected to electrophoresis in a 4%–12% NuPage Gel until the dye front reached the bottom of the gel cast. For cell and EV blots, equal amounts of protein content quantified by BCA assay were prepared in 1× NuPage Loading Buffer with BME and boiled for 5 min. Samples were loaded and subjected to electrophoresis in a 4%–12% NuPage Gel until the dye front reached the bottom of the gel cast. Prepared gels were placed in iBlot2 transfer stacks and transferred using the P0 setting on the iBlot 2 Gel Transfer Device. The PVDF membrane was blocked in TBS Odyssey Blocking buffer for 1 hr at RT. Membranes were washed in TBST and incubated with Streptavidin-800 (1:10,000 dilution, Licor, 926-32230) for 30 min in TBS Odyssey Blocking buffer +0.1% Tween 20. Membranes were washed in TBST 3× with a final wash in water. Membranes were visualized using an Odyssey DLx imager. Western blot samples were run and quantified 2–3 times and a representative image was displayed in figures.

For cell and EV blots, equal amounts of protein content quantified by BCA assay were prepared in 1× NuPage Loading Buffer with BME and boiled for 5 min. Samples were loaded and subjected to electrophoresis in a 4%–12% NuPage Gel until the dye front reached the bottom of the gel cast. Prepared gels were placed in iBlot2 transfer stacks and transferred using the P0 setting on the iBlot 2 Gel Transfer Device. The PVDF membrane was blocked in TBS Odyssey Blocking buffer for 1 hr at RT. Membranes were washed in TBST and incubated overnight in primary antibody at 4°C in TBS Odyssey Blocking buffer +0.1% Tween 20 while shaking. Primary antibodies used were ANPEP (R&D Systems, AF3815), FN1 (Abcam, ab2413), vimentin (Cell Signaling Technology, 5741S), ITIH4 (Atlas antibodies, HPA003948), MFGE8 (Thermo Fisher Scientific, PA5-82036), and IGSF8 (R&D Systems, AF3117-SP). Membranes were washed in 3× TBST before introduction to a 1:10,000 dilution of secondary antibody in TBS Odyssey Blocking buffer +0.1% Tween 20 for 1 hr at room temperature while shaking. Secondary antibodies used were Goat Anti-Rabbit HRP (Thermo Fisher Scientific, 31460) and Rabbit

Anti-Sheep HRP (Thermo Fisher Scientific, 31480). Blots were imaged after 5 min in the presence of SuperSignal West Pico PLUS Chemiluminescent Substrate (Thermo Fisher Scientific, 34577) and imaged using a ChemiDoc XRS+. Western blot samples were run and quantified 2–3 times and a representative image was displayed in figures. EV blot was run once due to limited sample availability. EVs derived for western blotting were cultured and harvested independently of either biological replicate used for mass spectrometry analysis.

## Proteomic preparation for whole EVs

Whole EV pellets were previously flash-frozen after collection. EVs were processed for LC-MS/MS using a PreOmics iST Kit (P.O.00027). Briefly, EV pellets were brought up in 50 µl of provided LYSE solution and boiled with agitation for 10 min. The provided enzymes mixture (Trypsin and LysC) were resuspended in 210 µl of RESUSPEND buffer, mixed, and added to the lysed EVs. Samples were allowed to mix at 500 rpm for 1.5 hr at 37°C, before being quenched with 100 µl of STOP solution. Sample was spun in provided C18 spin cartridge and washed 1× with 200 µl of WASH 1 and WASH 2. Peptides were eluted with 2× 100 µl of ELUTE, dried, and resuspended with the provided LC-LOAD solution. Peptides were quantified using Pierce Quantitative Colorimetric Peptide Assay (Thermo Fisher Scientific, 23275).

## Proteomic preparation for surface enriched samples

Frozen cell and EV pellets were lysed using 2× RIPA buffer (VWR) with protease inhibitor cocktail (Sigma-Aldrich; St. Louis, MO) at 4°C for 30 min. Cell lysate was then sonicated, clarified, and incubated with 100 µl of neutravidin agarose slurry (Thermo Fisher Scientific, 29204) at 4°C for 1 hr. The bound neutravidin beads were washed in a 2-ml Bio-spin column (Bio-Rad, 732-6008) with 5 ml RIPA buffer, 5 ml high salt buffer (1 M NaCl, PBS pH 7.5), and 5 ml urea buffer (2 M urea, 50 mM ammonium bicarbonate) to remove nonspecific proteins. Beads were allowed to fully drain before transferring to a Low-bind Eppendorf Tube (022431081) with 2 M Urea. Sample was spun down at 1000g and aspirated to remove excess liquid. Samples were brought up in 100 µl of 4 M Urea digestion buffer (50 mM Tris pH 8.5, 10 mM TCEP, 20 mM IAA, 4 M Urea) and allowed to reduce and alkylate for 10 min at 55°C while shaking. After the addition of 2 µg of total reconstituted Trypsin/LysC, the sample was incubated for 2 hr at RT. To activate the trypsin, mixture was diluted with 200 µl of 50 mM Tris pH 8.5 to a final Urea concentration of below 1.5 M. The mixture was covered and allowed to incubate overnight at RT. The mixture was isolated from the beads by centrifugation in a collection column (Pierce; 69725) before being acidified with 10% TFA until pH of 2.0 was reached. During this time, a Pierce C18 spin column (Pierce, 89873) was prepared as per manufacturing instructions. Briefly, C18 resin was washed twice with 200 µl of 50% LC-MS/MS grade ACN. The column was equilibrated with two 200 µl washes of 5% ACN/0.5% TFA. The pre-acidified sample was loaded into the C18 column and allowed to fully elute before washing two times with 200 µl washes of 5% ACN/0.5% TFA. One final wash of 200 µl 5% ACN/1% FA was done to remove any residual TFA from the elution. Samples were eluted in 70% ACN, dried, and dissolved in 0.1% formic acid, 2% acetonitrile prior to LC-MS/MS analysis. Peptides were quantified using Pierce Quantitative Colorimetric Peptide Assay (Thermo Fisher Scientific, 23275).

## LC-MS/MS

Liquid chromatography and mass spectrometry was performed as previously described (*Meier et al., 2020*). Briefly, approximately 200 ng of peptides were separate using a nanoElute UHPLC system (Bruker) with a pre-packed 25 cm × 75 µm Aurora Series UHPLC column+ CaptiveSpray insert (CSI) column (120 A pore size, IonOpticks, AUR2-25075C18A-CSI) and analyzed on a timsTOF Pro (Bruker) mass spectrometer. Peptides were separated using a linear gradient of 2%–34% solvent B (solvent A: 2% acetonitrile and 0.1% formic acid; solvent B: acetonitrile and 0.1% formic acid) over 100 min at 400 nl/min. Data-dependent acquisition was performed with parallel accumulation-serial fragmentation (PASEF) and trapped ion mobility spectrometry (TIMS) enabled with 10 PASEF scans per topN acquisition cycle. The TIMS analyzer was operated at a fixed duty cycle close to 100% using equal accumulation and ramp times of 100 ms each. Singly charged precursors were excluded by their position in the m/z–ion mobility plane, and precursors that reached a target value of 20,000 arbitrary units were dynamically excluded for 0.4 min. The quadrupole isolation width was set to 2 m/z for m/z < 700

and to 3 m/z for m/z > 700 and a mass scan range of 100–1700 m/z. TIMS elution voltages were calibrated linearly to obtain the reduced ion mobility coefficients (1/K0) using three Agilent ESI-L Tuning Mix ions (m/z 622, 922, and 1222).

## Data processing and analysis

Briefly, for general database searching, peptides for each individual dataset were searched using PEAKS Online X version 1.5 against both the PM annotated human proteome (Swiss-prot GOCC database, August 3, 2017 release) and the entire Swiss-prot Human Proteome (Swiss-prot). We acknowledge the identification of a number of proteins not traditionally annotated to the PM, which were published in the final Swiss-prot GOCC-PM database used. Additionally, to not miss any key surface markers such as secreted proteins or anchored proteins without a transmembrane domain, we chose to initially avoid searching with a more stringent protein list, such as the curated SURFY database. However, following the analysis, we bolded proteins found in the SURFY database and italicized proteins known to be secreted (Uniprot). Enzyme specificity was set to trypsin+ LysC with up to two missed cleavages. Cysteine carbamidomethylation was set as the only fixed modification; acetylation (N-term) and methionine oxidation were set as variable modifications. The precursor mass error tolerance was set to 20 PPM and the fragment mass error tolerance was set to 0.05 Da. Data was filtered at 1% for both protein and peptide FDR and triaged by removing proteins with fewer than two unique peptides. All mass spectrometry database searching was based on two biological replicates. Biological replicates underwent washing, labeling, and downstream LC-MS/MS preparation separately.

For comparative LFQ of cellular and EV samples, datasets were searched using PEAKS Online X version 1.5 against the PM annotated human proteome (Swiss-prot GOCC database, August 3, 2017 release). Enzyme specificity was set to trypsin+ LysC with up to two missed cleavages. Cysteine carbamidomethylation was set as the only fixed modification; acetylation (N-term) and methionine oxidation were set as variable modifications. The precursor mass error tolerance was set to 20 PPM and the fragment mass error tolerance was set to 0.05 Da. Data was filtered at 1% for both protein and peptide FDR and triaged by removing proteins with fewer than two unique peptides. Label-free quantification of protein was completed by taking the average intensity of the top three most intense peptides for each protein. Data was normalized by total area sum intensity for each sample. Using Perseus, all peak areas were log2(x) transformed and missing values were imputed separately for each sample using the standard settings (width of 0.3, downshift of 1.8). Significance was based on a standard unpaired Student t-test with unequal variances across all four replicates. Reported peak area values represent the averages of all four replicates—two biological and two technical replicates. For representation of the data in figures, a Z-score was computed and is defined as (LFQ Area−Mean LFQ Area)/Standard Deviation. Protein IDs that were not annotated to be secreted or expressed extracellularly were removed. Further, in the Cell versus EV graph, any proteins that showed a standard deviation (SD) greater than 1.5 between Control and Myc of each sample type (EV or Cell) were removed to avoid representation of oncogene-specific changes. Oncogene-specific changes are instead shown in *Figure 3—source data 3*, *Figure 4—source data 4*, and *Figure 5—source data 3*. EVs and cells from different biological replicates were cultured on different days. Desalting, quantification, and LC-MS/MS runs were performed together. The mass spectrometry proteomics data have been deposited to the ProteomeXchange Consortium via the PRIDE (*Perez-Riverol et al., 2019*) partner repository with the dataset identifier PXD028523.

To compare replicates of data in *Figure 3—figure supplement 2*, a simple linear regression was performed on total area sum intensity normalized data. Replicate one was graphed against Replicate two for biocytin hydrazide, NHS-biotin, and WGA-HRP, and the resulting data was shown with calculated R and p values as determined using the simple linear regression software suite in Prism. For the supplementary heatmap output, total area normalized LFQ data found in *Figure 5—source data 2* was loaded into Morpheus (software.broadinstitute.org/Morpheus) and data points were clustered by the Pearson correlation between all replicates on both columns and rows. This same data was used in Perseus to produce the PCA. Distinct gene clusters were further analyzed by functional annotation using the DAVID Bioinformatics Resource 6.8.

## Acknowledgements

The authors acknowledge the members of the Wells lab for their support. Special thanks to Alice Ting, PhD and Tess Branon, PhD for their helpful advice regarding enzyme purification. The authors would also like to acknowledge and thank the lab of Zev Gartner, PhD for providing materials and methods for DNA-lipid tethering. Additionally, the authors would like to thank Rushika Perera, PhD for donating the pancreatic cancer cell lines PaTu8902 and KP-4, which were used in this manuscript. JAW was supported by generous funding from the Chan Zuckerberg Biohub Investigator Program, the Harry and Dianna Hind Professorship, NIH (R35GM122451), and NCI (R01CA248323). The NIH F31 Ruth L Kirschstein National Research Service Award (1F31CA247527) supported LLK. The National Science Foundation Graduate Research Fellowship Program (1650113) supported SKE. RB and JY were supported by funding from the NIH (U01CA244452).

## Additional information

### Funding

| Funder | Grant reference number | Author |
|---|---|---|
| National Cancer Institute | 1F31CA247527 | Lisa L Kirkemo |
| National Science Foundation | 1650113 | Susanna Elledge |
| National Cancer Institute | U01CA244452 | Jiuling Yang<br>Robert Blelloch |
| National Institute of General Medical Sciences | R35GM122451 | James A Wells |
| National Cancer Institute | R01CA248323 | James A Wells |
| Chan Zuckerberg Biohub Investigator Program | | James A Wells |
| Harry and Dianna Hind Professorship | | James A Wells |

The funders had no role in study design, data collection and interpretation, or the decision to submit the work for publication.

### Author contributions

Lisa L Kirkemo, Susanna K Elledge, Conceptualization, Data curation, Formal analysis, Investigation, Methodology, Project administration, Validation, Visualization, Writing – original draft, Writing – review and editing; Jiuling Yang, Resources; James R Byrnes, Methodology, Writing – review and editing; Jeff E Glasgow, Conceptualization, Writing – review and editing; Robert Blelloch, Resources, Supervision, Writing – review and editing; James A Wells, Funding acquisition, Resources, Supervision, Writing – review and editing

### Author ORCIDs
Lisa L Kirkemo http://orcid.org/0000-0003-1686-6987
Susanna K Elledge http://orcid.org/0000-0002-9621-3881
James R Byrnes http://orcid.org/0000-0003-0297-1209
Robert Blelloch http://orcid.org/0000-0002-1975-0798
James A Wells http://orcid.org/0000-0001-8267-5519

### Decision letter and Author response
Decision letter https://doi.org/10.7554/eLife.73982.sa1
Author response https://doi.org/10.7554/eLife.73982.sa2

## Additional files

### Supplementary files
• Transparent reporting form

• Source data 1. Raw and Uncropped Western Blots.

## Data availability

All data has been deposited alongside the manuscript as supporting source data files. Raw western blot images are made available as a source file titled "Raw_WesternBlot". All data from mass spectrometry experiments are provided as source data within the manuscript. "Figure 3-source data 2" details the quantification results from the RWPE-1 +/- Myc cell comparison experiments. "Figure 4-source data 2" details the quantification results from the RWPE-1 +/- Myc EV comparison experiments. "Figure 4-source data 3" details the quantification results from the RWPE-1 +/- Myc whole EV experiments. "Figure 5-source data 2" details the quantification results from PEAKS and Perseus for the RWPE-1 +/- Myc EV and cell comparison experiments. "PaTu8902_WGAvsAPEX2_DatabaseSearch" and "KP4_APEX_HRP_Comparison_DatabaseSearch" documents detail results from APEX2 and HRP method comparisons across two different PDAC cell lines. "RWPE_Method_Comparison_DatabaseSearch" outlines the results from the NHS-biotin, biocytin hydrazide, and WGA-HRP comparison experiments performed on RWPE EV and Myc transduced cells. The mass spectrometry proteomics data have been deposited to the ProteomeXchange Consortium via the PRIDE partner repository with the dataset identifier PXD028523.

The following dataset was generated:

| Author(s) | Year | Dataset title | Dataset URL | Database and Identifier |
|---|---|---|---|---|
| Kirkemo LL, Yang J, Byrnes J, Glasgow J, Blelloch R, Wells JA, Elledge SK | 2021 | Cell-surface tethered promiscuous biotinylators enable comparative small-scale surface proteomic analysis of human extracellular vesicles and cells | https://www.ebi.ac.uk/pride/archive/projects/PXD028523 | PRIDE, PXD028523 |

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
