## [Editor Report]

This work presents useful technical options in the detection and comparison of the cell surface and extracellular vesicle (EV) proteomes of normal and myc-transformed cells in culture. The procedures and the proteomes identified show promise of wider application in clinical and basic research settings. The work highlights distinctions in the two proteomes and the influence of myc-transformation on the selection of cargo molecules for capture in EVs secreted by human cells in vitro.

---

## [Decision Letter]

**Decision letter after peer review:**

Thank you for submitting your article "Cell-surface tethered promiscuous biotinylators enable small-scale surface proteomics of human exosomes" for consideration by *eLife*. Your article has been reviewed by 3 peer reviewers, including Randy Schekman as the Reviewing Editor and Reviewer #1, and the evaluation has been overseen by Suzanne Pfeffer as the Senior Editor. The following individuals involved in review of your submission have agreed to reveal their identity: David R Walt (Reviewer #2); Clotilde Thery (reviewer #3)

Essential revisions:

1) All reviewers agree that your use of the term exosome must be replaced by "extracellular vesicles" or EVs. The procedure you have used does not purify exosomes but instead a crude collection of sedimentable particles or possibly vesicles depending on whether you used sucrose flotation, which as discussed by reviewer #1 was not clear in your Methods section.

2) Reviewers #2 and 3 are quite critical of the proteomic analysis you have conducted. #2 argues that your proteome may well include proteins bound to the cell surface but secreted by other cells. #3 is more critical and argues that your analysis is not much of an improvement over other work that has already been published. The reviewer points out that some of the proteins you have identified are not actually exposed on the surface of the cell or of the EV. Further, #3 requests that you document the value of the technique for small-scale EV analysis, hence, to provide detailed technical info on the amount of EVs (number of source cells, and/or amount of proteins or particles) required to perform the surface proteome. If this amount does not really qualify as "small-scale", the authors must revise their message (title).

3) Reviewer #1 is most critical of the description of and procedures you used to obtain the EV fraction. #1 suggests that you have missed an opportunity to distinguish the surface proteome of crude EVs in relation to purified exosomes.

We believe this work could be published in *eLife*, but not until you have completed a significant additional analysis of the proteome data to better justify your conclusions. Should these improvements, particularly those noted by reviewer #3, not be possible to complete, you may wish to consider the approach recommended by reviewer #1 where you resolve exosomes from the crude EV fraction to identify proteins that sort selectively into one vesicle sub-type.

*Reviewer #1 (Recommendations for the authors):*

The techniques and results of this manuscript appear novel and likely an improvement over other, non-adherent, labeling cell surface labeling methods. The results extend to extracellular vesicles and suggest that some proteins are highly enriched and may be diagnostic of a tumor cell origin. As such, the work represents a useful advance in the field and may merit publication as a technical report.

I was less convinced by the approach taken to characterize what the authors refer to as exosomes. According to their methods section, the particulate material they collect is the result of an simple enrichment by differential centrifugation. If so, the material is a crude mixture of various vesicles and sedimentable particles, not necessarily all of membrane origin. To add confusion to this method, the authors use a diagram in Figure 4A that refers to membranes that are floated on a sucrose gradient and reference the work of Poggio et al. (2019) who reported the use of a 20-60% sucrose gradient to isolate exosomes. However, in that report, the gradient is not shown and the relevant fraction is referred to as a sucrose light fraction containing the exosome marker CD63. It is not clear from the Poggio et al. work if they have resolved exosomes from other buoyant membrane vesicles such as those that are shed by budding from the cell surface. The current work is even less clear on that distinction. In work published since the Poggio et al. paper, two labs have described the use of an Optiprep buoyant density gradient to resolve membranes into two subtypes with a higher buoyant density fraction likely representing authentic exosomes (Jeppesen et al., 2019, Cell 177, 428-445 April 4, 2019. https://doi.org/10.1016/j.cell.2019.02.029 ;Temoche-Diaz et al. 2019 https://doi.org/10.7554/*eLife*.47544.001). As an example, in the current work, the authors describe MFGE8 as a "pan exosomal marker" whereas in Temoche-Diaz et al., this protein is seen to be nearly exclusively in a light buoyant density fraction likely corresponding to cell surface budded vesicles and not exosomes.

There are two remedies to the problem of use of the term exosomes in this manuscript. The simplest would be to tighten up the description of the isolation method and switch the term to extracellular vesicles. And even then, it will be important to know if the procedure is as described by Poggio et al. with the use of a buoyant density centrifugation step. However, if this work really did rely on crude differential centrifugation, then there should be some additional experimental effort to demonstrate that the biotinylated proteins are from a membrane as opposed to a sedimentable particle. The two publications cited above demonstrate that not all sediments along with extracellular vesicles is actually a membrane.

Another remedy to the misuse of the term exosomes would be to conduct a more refined fractionation and then to compare the surface proteomes of EV sub-fractions.

*Reviewer #2 (Recommendations for the authors):*

1. Explain how one would deal with non-specifically bound proteins.

2. Explain how one would use the method for time-sensitive changes.

3. Verify that exosomes is correct terminology or replace with extracellular vesicles.

*Reviewer #3 (Recommendations for the authors):*

Of note, the authors should replace the term "exosomes" by the generic term "extracellular vesicles" throughout their article (title, abstract, text), since their method of EV isolation, even if it includes a density gradient step, co-isolates EVs originating from the plasma membrane and from endosomes which display the same density (see Witwer and Thery, J Extracell Vesicles 2019 # 31489144 for explanation on nomenclature). Consequently, the abbreviation "EV" for the control RWPE1 cells should be changed.

Experimental details are missing, for instance for the Western blots: what does "equal amounts of samples" mean? Quantified by protein content, or by number of cells or of EV-producing cells, or by biotinylation level? For the density gradient: are the EV pellets after biotinylation loaded at the bottom or on top of the gradient and in which fractions are the EVs collected? Are the other fractions also analysed at least for presence of biotinylated proteins by global Western blot? In general, what amount of EVs and from how many cells must be used to obtain interpretable surface proteomic results.

[Editors' note: further revisions were suggested prior to acceptance, as described below.]

Thank you for resubmitting your article "Cell-surface tethered promiscuous biotinylators enable comparative small-scale surface proteomic analysis of human extracellular vesicles and cells" for consideration by *eLife*. Your revised article has been reviewed by 3 peer reviewers, including Randy Schekman as the Reviewing Editor and Reviewer #1, and the evaluation has been overseen by Suzanne Pfeffer as the Senior Editor. The following individuals involved in review of your submission have agreed to reveal their identity: David R Walt (Reviewer #2); Clotilde Thery (Reviewer #3).

Essential revisions:

This report describes a new technique to detect the surface proteome of normal and myc-transformed cells in relation to extracellular vesicles from the same cells. The data obtained from this comparison may be useful in evaluating cell surface and extracellular vesicle marker proteins that may be of diagnostic value. The article could possibly be more useful with some small additional modifications to the text.

Reviewers #1 and 2 are fine with this version of the manuscript however #2 has some more recommended changes. Please consider these additional changes and return a final version of the manuscript which I will review without further consultation.

*Reviewer #1 (Recommendations for the authors):*

The authors responded appropriately to my main concerns. They changed the nomenclature to refer to their vesicles as EVs rather than exosomes, and clarified the use of buoyant density centrifugation of EVs to ensure their preparation is bona fide membrane material rather than just sedimentable particles. They chose not to perform the additional separation needed to examine exosomes as well as total EVs. I understand and accept this although it seems unlikely that they will follow through to do that work, instead leaving it for others in the field to complete that analysis.

I look forward to a discussion with the other reviewers to assess their evaluation of the rest of the revisions on this manuscript.

*Reviewer #2 (Recommendations for the authors):*

The authors have responded to all my comments and suggestions. They have carried out additional experiments and reanalyzed their data. They have added more details to the Methods section.

*Reviewer #3 (Recommendations for the authors):*

The revised version of the article properly addresses the previous comments of reviewers.

I appreciate the additional data shown in figures 3 and 4, and new presentation of analyses in figure 5.

I would suggest to revise the scheme of EV isolation now depicted in figure 4A (formerly shown as suppl figure), as it is not entirely clear or correct:

"multivesicular bodies" above the arrow of centrifugation at 12,000g is probably a mistake (MVBs should not be recovered from the conditioned medium, the authors probably meant medium/large EVs?).

In the "sucrose gradient" part, indicate time of the 100,000g centrifugations, clarify that the fraction at 20-40% sucrose, ie around the middle of the gradient is recovered and analysed maybe after dilution in PBS before further centrifugation (currently it looks like all 6 fractions are recovered, unclear how they are treated, it looks like another centrifugation into a gradient 20_40%).

In Figure 3D and 4D, I am surprised that vimentin (VIM), a cytoskeletal protein, is bolded ie listed as identified by the SURFY database as having a transmembrane domain. I cannot see any indication of such domain in the Uniprot database, and when searching for vimentin in the surfy database (http://wlab.ethz.ch/surfaceome/), no protein was retrieved. Other proteins listed in figure 4 was also not retrieved (JUP, a junction plakoglobin internal component, VCL = vinculin) The authors should double check their listing and verify the explanation on how these proteins were specifically selected for analysis.

A detail on the description of results of figure4B, and the MVB origin of the small EVs analysed here: according to Mathieu et al. 2021 (quoted by the authors), CD9 and probably CD81 are not specific of MVB-derived EVs, whereas CD63, LAMP2 (probably LAMP1), syntenin are probably more likely markers of exosomes.

Sorry for not highlighting it in the previous review, but the authors could usefully compare (as a sentence or short paragraph in the discussion) their data with a recent publication showing an effect of several oncogenes on EV release, including Myc, which was shown to increase EV release and change the global protein composition: Kilinc et al., Dev Cell 2021, DOI 10.1016/j.devcel.2021.05.014. In particular in M&M: can the authors indicate if more EVs were recovered per Myc cell than per wt cell (as observed by Kilinc et al)? Currently, the only detail is that twice less Myc cells were seeded initially than wt, but maybe this results in the same number at time of EV collection, if Myc cells divide faster.

---

## [Author Response]

Essential revisions:1) All reviewers agree that your use of the term exosome must be replaced by "extracellular vesicles" or EVs. The procedure you have used does not purify exosomes but instead a crude collection of sedimentable particles or possibly vesicles depending on whether you used sucrose flotation, which as discussed by reviewer #1 was not clear in your Methods section.

We appreciate the reviewers for bringing this distinction to our attention. We have revised our manuscript and replaced all mentions of “exosome” with “small extracellular vesicles” (EVs). We have additionally updated our Methods section with the text below to clarify that we did perform a sucrose flotation that would remove sedimentable particles.

“RWPE-1 Control and Myc cells were plated at 7 million and 4 million cells per plate, respectively, across 16 x 15 cm^2^ plates and allowed to grow in normal keratinocyte-SFM media with provided supplements. Small EVs were isolated as previously described. Briefly, two days prior to EV isolation, media was replaced with 15 milliliters BPE-free keratinocyte-SFM media. For vesicle enrichment, media was isolated after two days in BPE-free media and centrifuged at 300 x g for 10 minutes at RT, followed by 2,000 x g for 20 minutes at 4°C. Large debris was cleared by a 12,000 x g spin for 40 minutes at 4°C. The pre-cleared supernatant was spun a final time at 100,000 x g at 4°C for 1 hr to pellet extracellular vesicles. Isolated extracellular vesicles were brought up in 50 µl of PBS with 0.5 µM of WGA-HRP and the mixture was allowed to bind on ice for 5 minutes. WGA-HRP bound vesicles were placed on a shaker (500 rpm) at 37°C before the addition of biotin tyramide (0.5 mM final concentration) and H2O2 (1 mM final concentration). Vesicles underwent labeling for 2 minutes before being quenched with 5 mM Trolox /10 mM Sodium Ascorbate /1 mM Sodium Pyruvate. Biotinylated small EVs were purified from other sedimentable particles by further centrifugation on a sucrose gradient (20-60%) for 16 hours at 4°C at 100,000xg. Precisely, the gradient was loaded using 0%, 20%, 40%, and 60% sucrose fractions from top to bottom. The sample was loaded at the bottom in 60% sucrose and the purified small EVs were isolated in the 20-40% sucrose fractions. Differential sucrose centrifugation generally yielded between 3-5 µg of small EVs from both RWPE-1 Control and Myc samples.”

2) Reviewers #2 and 3 are quite critical of the proteomic analysis you have conducted. #2 argues that your proteome may well include proteins bound to the cell surface but secreted by other cells. #3 is more critical and argues that your analysis is not much of an improvement over other work that has already been published. The reviewer points out that some of the proteins you have identified are not actually exposed on the surface of the cell or of the EV. Further, #3 requests that you document the value of the technique for small-scale EV analysis, hence, to provide detailed technical info on the amount of EVs (number of source cells, and/or amount of proteins or particles) required to perform the surface proteome. If this amount does not really qualify as "small-scale", the authors must revise their message (title).

We thank the reviewers for the comments to strengthen our proteomic analysis. We believe that any low affinity, non-specific binding proteins are likely removed in the multiple wash/centrifugation steps on cells or the multiple centrifugation steps and sucrose gradient purification on EVs. As such, we believe that the secreted proteins identified are likely high affinity interactions and their differential expression on either cells or EVs play an important part in the downstream biology of both sample types. Proteins identified in our LC-MS/MS analysis are all annotated surface associated proteins or secreted proteins found in the Uniprot GOCC (Gene Ontology Cellular Component) Plasma Membrane database (5,746 annotated surface proteins in the database). The GOCC database is the most inclusive list of membrane proteins, only requiring they contain either a signal sequence, transmembrane domain, GPI anchor or other membrane associated motifs yielding a total of 5,746 proteins. This will include organelle membrane proteins. It is known that proteins can traffic from intracellular organelles to the cell surface, so these can be bonified cell surface proteins too. However, to increase stringency in our analysis, we have updated our figures to exclude intracellular membrane localized proteins, which are included in the Uniprot GOCC database. For further clarity in the data presentation (Figure 3D, 4D, 5D and corresponding supplementary lists), we have presented only proteins either found in the SURFY database (shown in bolded text) that only includes surface annotated proteins with a predicted transmembrane domain (Bausch-Fluck et al., The in silico human surfaceome. *PNAS*. 2018) or proteins that are annotated to be secreted from the cell to the extracellular space (Uniprot classification, shown in *italic* text).

We have updated the manuscript to describe the amount of EV sample prepared to demonstrate small-scale analysis. While we had to use 16x15cm^2^ plates (240 milliliters of total media) to isolate EVs—due to the low production rate of EVs from the RWPE cell line—the total amount of isolated material post-sucrose gradient was 3-5 µg of material, which we believe to be small-scale.

3) Reviewer #1 is most critical of the description of and procedures you used to obtain the EV fraction. #1 suggests that you have missed an opportunity to distinguish the surface proteome of crude EVs in relation to purified exosomes.

We appreciate the reviewer’s suggestion and agree that this distinction would be an interesting application of this method. While we are interested in performing this experiment, the goal of our initial proof-of-concept study was to focus on the comparative analysis of extracellular vesicle surface proteins to parent cell surface proteins. Therefore, we believe the suggested experiment would be better suited for a follow up paper on this method.

We believe this work could be published in eLife, but not until you have completed a significant additional analysis of the proteome data to better justify your conclusions. Should these improvements, particularly those noted by reviewer #3, not be possible to complete, you may wish to consider the approach recommended by reviewer #1 where you resolve exosomes from the crude EV fraction to identify proteins that sort selectively into one vesicle sub-type.Reviewer #1 (Recommendations for the authors):The techniques and results of this manuscript appear novel and likely an improvement over other, non-adherent, labeling cell surface labeling methods. The results extend to extracellular vesicles and suggest that some proteins are highly enriched and may be diagnostic of a tumor cell origin. As such, the work represents a useful advance in the field and may merit publication as a technical report.

We appreciate the reviewer’s summary and for recognizing the merit and important advance of this work.

I was less convinced by the approach taken to characterize what the authors refer to as exosomes. According to their methods section, the particulate material they collect is the result of an simple enrichment by differential centrifugation. If so, the material is a crude mixture of various vesicles and sedimentable particles, not necessarily all of membrane origin. To add confusion to this method, the authors use a diagram in Figure 4A that refers to membranes that are floated on a sucrose gradient and reference the work of Poggio et al. (2019) who reported the use of a 20-60% sucrose gradient to isolate exosomes. However, in that report, the gradient is not shown and the relevant fraction is referred to as a sucrose light fraction containing the exosome marker CD63. It is not clear from the Poggio et al. work if they have resolved exosomes from other buoyant membrane vesicles such as those that are shed by budding from the cell surface. The current work is even less clear on that distinction. In work published since the Poggio et al. paper, two labs have described the use of an Optiprep buoyant density gradient to resolve membranes into two subtypes with a higher buoyant density fraction likely representing authentic exosomes (Jeppesen et al., 2019, Cell 177, 428-445 April 4, 2019. https://doi.org/10.1016/j.cell.2019.02.029 ;Temoche-Diaz et al. 2019 https://doi.org/10.7554/eLife.47544.001). As an example, in the current work, the authors describe MFGE8 as a "pan exosomal marker" whereas in Temoche-Diaz et al., this protein is seen to be nearly exclusively in a light buoyant density fraction likely corresponding to cell surface budded vesicles and not exosomes.There are two remedies to the problem of use of the term exosomes in this manuscript. The simplest would be to tighten up the description of the isolation method and switch the term to extracellular vesicles. And even then, it will be important to know if the procedure is as described by Poggio et al. with the use of a buoyant density centrifugation step. However, if this work really did rely on crude differential centrifugation, then there should be some additional experimental effort to demonstrate that the biotinylated proteins are from a membrane as opposed to a sedimentable particle. The two publications cited above demonstrate that not all sediments along with extracellular vesicles is actually a membrane.Another remedy to the misuse of the term exosomes would be to conduct a more refined fractionation and then to compare the surface proteomes of EV sub-fractions.

We apologize for the confusion. We did indeed use the same sucrose gradient approach for proteomic analysis as described in Poggio et al., which enriches for exosomes based on density. In the Poggio paper, there were several levels of characterization of the small EVs isolated by this method, including dependency on Rab27a and nSMase2, co-fractionation with exosome markers (CD63, HRS), and electron microscopy – (Figures 1 and 2 in Poggio et al). However, since we did not purify exosomes away from surface budded vesicles, we have changed all mentions of exosome to small extracellular vesicles. We have also edited the methods section to further clarify the use of the sucrose gradient. Furthermore, we have run an additional whole exosome lysate experiment to confirm that our purification approach strongly enriches for known markers of MVB-derived extracellular vesicles, such as CD63, CD81, CD9, SDCB1, LAMP1, LAMP2, and ALIX. This is shown in the waterfall plot, which has been added to the manuscript as Figure 4B.

Reviewer #2 (Recommendations for the authors):1. Explain how one would deal with non-specifically bound proteins.

We utilized the most rigorous informatics analysis available (Uniprot and SURFY) to annotate the proteins we find as having a signal sequence and/or TM domain. Data shown in heatmaps are based off of significance (p < 0.05) across all four replicates, which supports that any secreted proteins present are likely due to actual biological differences between oncogenic status and/or sample origin (i.e. EV vs cell).

2. Explain how one would use the method for time-sensitive changes.

We thank the reviewer for this comment and giving us an opportunity to elaborate on the types of experiments enabled by this new method. A previous study (Y, Li et al. Rapid Enzyme-Mediated Biotinylation for Cell Surface Proteome Profiling. *Anal. Chem.* 2021) showed that labeling the cell surface with soluble HRP allowed the researchers to detect immediate surface protein changes in response to insulin treatment. They demonstrated differential surfaceome profiling changes at 5 minutes vs 2 hours following treatment with insulin. Only methods utilizing these rapid labeling enzymes could allow for this type of resolution. A few other biological settings that experience rapid cell surface changes are: response to drug treatment, T-cell activation and synapse formation (S, Valitutti, et al. The space and time frames of T cell activation at the immunological synapse. *FEBS Letters.* 2010) and GPCR activation (T, Gupte et al. Minute-scale persistence of a GPCR conformation state triggered by non-cognate G protein interactions primes signaling. Nat. Commun. 2019). We also believe the method would be useful for post-translational processes where proteins are rapidly shuttling to the cell surface. We have updated the discussion to elaborate on these types of experiments.

“Due to the fast kinetics of peroxidase enzymes (1-2 min), our approaches could enable kinetic experiments to capture rapid post-translational trafficking of surfaces proteins, such as response to insulin, certain drug treatments, T-cell activation and synapse formation, and GPCR activation.”

3. Verify that exosomes is correct terminology or replace with extracellular vesicles.

We have systematically changed from using exosomes to small extracellular vesicles and better defined the isolation procedure that we used in the methods section.

Reviewer #3 (Recommendations for the authors):Of note, the authors should replace the term "exosomes" by the generic term "extracellular vesicles" throughout their article (title, abstract, text), since their method of EV isolation, even if it includes a density gradient step, co-isolates EVs originating from the plasma membrane and from endosomes which display the same density (see Witwer and Thery, J Extracell Vesicles 2019 # 31489144 for explanation on nomenclature). Consequently, the abbreviation "EV" for the control RWPE1 cells should be changed.

We thank the reviewer for this comment and have updated both the term exosomes to extracellular vesicles and the EV (empty vector) to control.

Experimental details are missing, for instance for the Western blots: what does "equal amounts of samples" mean? Quantified by protein content, or by number of cells or of EV-producing cells, or by biotinylation level? For the density gradient: are the EV pellets after biotinylation loaded at the bottom or on top of the gradient and in which fractions are the EVs collected? Are the other fractions also analysed at least for presence of biotinylated proteins by global Western blot? In general, what amount of EVs and from how many cells must be used to obtain interpretable surface proteomic results.

We thank the reviewer for highlighting this area where we could provide more methodological detail. We have updated the methods and have addressed this. We did not analyze the other fractions by western blot for biotinylated proteins in this study. For comparison between cells and EVs, total protein content was used to normalize loading amounts for western blotting. The methods section has been amended as shown below:

“For cell and EV blots, equal amounts of protein content quantified by BCA assay were prepared in 1X NuPage Loading Buffer with BME and boiled for 5 minutes.”

[Editors' note: further revisions were suggested prior to acceptance, as described below.]

Essential revisions:This report describes a new technique to detect the surface proteome of normal and myc-transformed cells in relation to extracellular vesicles from the same cells. The data obtained from this comparison may be useful in evaluating cell surface and extracellular vesicle marker proteins that may be of diagnostic value. The article could possibly be more useful with some small additional modifications to the text.Reviewers #1 and 2 are fine with this version of the manuscript however #2 has some more recommended changes. Please consider these additional changes and return a final version of the manuscript which I will review without further consultation.Reviewer #3 (Recommendations for the authors):The revised version of the article properly addresses the previous comments of reviewers.I appreciate the additional data shown in figures 3 and 4, and new presentation of analyses in figure 5.I would suggest to revise the scheme of EV isolation now depicted in figure 4A (formerly shown as suppl figure), as it is not entirely clear or correct:"multivesicular bodies" above the arrow of centrifugation at 12,000g is probably a mistake (MVBs should not be recovered from the conditioned medium, the authors probably meant medium/large EVs?).In the "sucrose gradient" part, indicate time of the 100,000g centrifugations, clarify that the fraction at 20-40% sucrose, ie around the middle of the gradient is recovered and analysed maybe after dilution in PBS before further centrifugation (currently it looks like all 6 fractions are recovered, unclear how they are treated, it looks like another centrifugation into a gradient 20_40%).

We are happy to further clarify the procedure for generating our EVs and have done so in Figure 4A as suggested.

In Figure 3D and 4D, I am surprised that vimentin (VIM), a cytoskeletal protein, is bolded ie listed as identified by the SURFY database as having a transmembrane domain. I cannot see any indication of such domain in the Uniprot database, and when searching for vimentin in the surfy database (http://wlab.ethz.ch/surfaceome/), no protein was retrieved. Other proteins listed in figure 4 was also not retrieved (JUP, a junction plakoglobin internal component, VCL = vinculin) The authors should double check their listing and verify the explanation on how these proteins were specifically selected for analysis.

We appreciate the thorough nature of the feedback and have added additional text to clarify. In particular, vimentin has been empirically identified at the extracellular interface of the plasma membrane, especially in the context of cancer (https://pubmed.ncbi.nlm.nih.gov/25487874/, https://www.ncbi.nlm.nih.gov/pmc/articles/PMC5342141/). For clarity of data presentation, we have removed the bold font, indicating SURFY inclusion. Likewise, JUP and VCL are both listed as plasma membrane proteins in Uniprot and Protein Atlas, taking part in cell-cell, as well as cell-matrix interactions. To maintain the highest level of rigor and clarity in our data presentation, we have removed JUP and VCL, and have added text to support the addition of vimentin.

While vimentin has traditionally been described as an intracellular protein, an extracellular membrane-bound form has been found to be important in the context of cancer (10.1002/ijc.29382, 10.18632/oncotarget.12458).

A detail on the description of results of figure4B, and the MVB origin of the small EVs analysed here: according to Mathieu et al. 2021 (quoted by the authors), CD9 and probably CD81 are not specific of MVB-derived EVs, whereas CD63, LAMP2 (probably LAMP1), syntenin are probably more likely markers of exosomes.Sorry for not highlighting it in the previous review, but the authors could usefully compare (as a sentence or short paragraph in the discussion) their data with a recent publication showing an effect of several oncogenes on EV release, including Myc, which was shown to increase EV release and change the global protein composition: Kilinc et al., Dev Cell 2021, DOI 10.1016/j.devcel.2021.05.014. In particular in M&M: can the authors indicate if more EVs were recovered per Myc cell than per wt cell (as observed by Kilinc et al)? Currently, the only detail is that twice less Myc cells were seeded initially than wt, but maybe this results in the same number at time of EV collection, if Myc cells divide faster.

We appreciate Dr. Thery’s comment and believe that a direct comparison could be very interesting. Indeed, we also found an increased production of EVs in the context of Myc overexpression, even after accounting for rapid cell growth in the Myc cells. We have included a short excerpt in the text to showcase the parallels between our data and those found by Kilinc et al.

After normalizing for cell number, we found the Myc cells produced nearly 40% more EVs than the corresponding control cells, which is consistent with previous work that has shown Myc overexpression yields higher quantities of EVs (10.1016/j.devcel.2021.05.014).